# CocoRNA: Collective RNA Design with Cooperative Multi-agent Reinforcement Learning

## Abstract

Designing RNA sequences that reliably fold into specific secondary structures is essential for understanding their biological functions but remains a challenging computational problem. We propose CocoRNA, a cooperative multi-agent reinforcement learning framework for RNA inverse design. CocoRNA simplifies the design task by decomposing it into smaller sub-problems, each solved collaboratively by multiple agents. This approach reduces the complexity of the problem and improves the exploration of design policies. During training, a centralized critic uses global structural information to guide the agents, enabling them to jointly optimize their design strategies. As a result, CocoRNA learns high-quality RNA design policies that generalize effectively to unseen structures without additional training. Experiments on the `Rfam` dataset demonstrate that CocoRNA substantially outperforms state-of-the-art methods in both success rate and design speed. Further experiments on other biological sequence design tasks highlight the effectiveness and broad potential of CocoRNA for complex design tasks. Visualization examples are available on our supplemental website.

## 1 Introduction

RNA is a polymeric molecule composed of nucleotides. Each nucleotide contains one of four bases: adenine (A), guanine (G), cytosine (C), or uracil (U). These bases carry genetic information. Nucleotides fold into complex secondary structures following Watson-Crick and Wobble base-pairing rules. Common structural elements include stems, loops, bulges, and junctions. For RNA, secondary structures are often more critical for function than they are in proteins (Kwon, 2025). These structures participate directly in catalysis, ligand recognition, and molecular scaffolding across diverse RNA classes (Vandivier et al., 2016; Bose et al., 2024). Examples include ribosomal RNAs (Ramakrishnan, 2014), messenger RNAs (Rouskin et al., 2014), and small nuclear RNAs (Fica et al., 2013). Secondary structures also regulate translation initiation and influence RNA polymerase II activity, thus controlling gene translation and splicing (Georgakopoulos-Soares et al., 2022). However, the precise roles of RNA secondary structures in many biological processes remain unclear (Georgakopoulos-Soares et al., 2022). To fully understand RNA functions, researchers need reliable methods to investigate and manipulate these structures (Vandivier et al., 2016). Computational inverse design can address this need, guiding experiments and reducing costs associated with testing structural hypotheses (Dotu et al., 2013; Morandi et al., 2022).

Computational RNA secondary structure design is essentially an inverse problem. The goal is to find RNA sequences that fold into a specific secondary structure with minimum free energy. This problem was first introduced in the early 1990s (Hofacker et al., 1994b). It is mathematically ill-posed in the Hadamard sense because multiple distinct sequences can fold into the same structure, as illustrated in Figure 1. Solving this inverse design problem is critical for advancing functional RNA research (Taft et al., 2010; Serganov & Nudler, 2013). Key applications include synthetic RNA biology (Dykstra et al., 2022) and RNA nanostructure design (Guo, 2010).

A major challenge in computational RNA design is the exponential growth of the sequence space. For an RNA sequence of length $\ell$, there are $4^\ell$ possible sequences. However, only a small fraction of these sequences will fold into the desired structure. For example, a 74-nucleotide tRNA with $40$ paired and $34$ unpaired positions results in a search space of approximately $10^{36}$ sequences. This enormous search space makes brute-force approaches computationally infeasible.

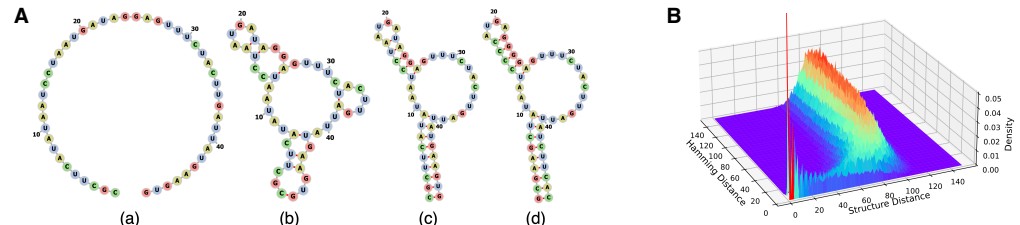

Figure 1: **A**. Examples of RNA sequences and structures visualized using Forna (Kerpedjiev et al., 2015). $(a)$ is an unfolded RNA sequence. Structures $(b)$ and $(c)$ show two different folded forms of this sequence. Structure $(c)$ is specifically generated by RNAfold (Lorenz et al., 2011), which uses a minimum free energy algorithm. Structure $(d)$ is folded from a different sequence, yet it shares the same secondary structure as $(c)$. **B**. Structure density surface for RNA sequences of length 150. The plot illustrates the distribution of Hamming distances between RNA sequences and their corresponding secondary structure distances.

Stochastic optimization is a widely used approach for computational RNA design. These methods typically rely on RNA folding prediction tools, such as ViennaRNA's RNAfold (Lorenz et al., 2011), mfold/UNAFold (Markham & Zuker, 2008), and RNAstructure (Ali et al., 2023). These tools predict RNA structures by minimizing free energy according to established thermodynamic parameters (Mathews et al., 1999; 2004). The design process begins with an initial seed sequence. Techniques such as adaptive random walks (Hofacker et al., 1994a) or stochastic local search iteratively mutate and evaluate candidate sequences until a suitable solution emerges. Well-known algorithms in this domain are RNA-SSD (Andronescu et al., 2004a) and INFO-RNA (Busch & Backofen, 2006a). Other effective strategies draw from evolutionary algorithms (Taneda, 2012; Esmaili-Taheri et al., 2014; Esmaili-Taheri & Ganjtabesh, 2015), constraint programming (Garcia-Martin et al., 2013; 2015), ant colony optimization (Kleinkauf et al., 2015), and ensemble optimization (Zhou et al., 2023).

Neural combinatorial optimization (NCO) has recently emerged as a powerful approach for combinatorial problems. Examples include the traveling salesman problem (Bello et al., 2016), maximum cut (Barrett et al., 2020), and bin packing (Jiang et al., 2021). NCO combines the representational strength of deep neural networks with the search efficiency of reinforcement learning (RL). In computational RNA design, Eastman et al. (2018) first introduced a pre-trained RL policy. Their policy learned RNA folding rules to guide the search process. Experiments on the Eterna100 benchmark (Anderson-Lee et al., 2016) demonstrated encouraging performance compared to stochastic optimization methods. Later, Runge et al. (2019) proposed LEARNA, an RL-based approach that directly generates RNA sequences, enhancing success rates and computational efficiency.

Despite these advances, RL-based RNA inverse design still faces three major challenges:

- 🚗 **Combinatorial explosion and sample inefficiency:** The RNA design space grows exponentially with sequence length, requiring enormous sample sizes. As a result, RL models struggle to accurately estimate value functions from limited data.
- 🚗 **Sparse rewards and local optima:** Very few sequences exactly match the target structure, creating sparse rewards. Additionally, the RNA design landscape contains numerous local optima (see Figure 1). This combination makes policy improvement challenging.
- 🚗 **Delayed rewards and misleading auxiliaries:** A non-zero reward in RNA inverse design only occurs when a correct sequence is found. Random exploration rarely finds these sequences, causing most attempts to yield no useful feedback. Long task horizons further complicate intermediate evaluations. Some methods (Angermueller et al., 2019; Runge et al., 2019) use auxiliary rewards to guide exploration. However, these signals can overshadow the primary objective in long tasks, causing policies to overfit short-term gains and become trapped in local optima.

These challenges significantly limit the effectiveness of current RL methods (Eastman et al., 2018; Runge et al., 2019). As demonstrated by our experiments in Section 4, existing approaches typically require hours of computation and tens of thousands of search steps, yet achieve only modest success rates. Furthermore, once algorithms become trapped in local optima, additional search efforts yield diminishing returns.

> 💡 *Conceptually, navigating the RNA sequence space resembles exploring a complex maze. Multiple explorers, each starting from different points and sharing their discoveries, can map this maze more efficiently than a single explorer. Inspired by this analogy, we propose using multi-agent reinforcement learning (MARL) to tackle computational RNA design.*

- 💡 We formulate computational RNA design as a collective task, where multiple agents cooperate to generate RNA sequences. To our knowledge, this is the first study to frame biological sequence design explicitly as a multi-agent decision-making problem and the first to apply MARL in this domain.
- 💡 We develop CoCoRNA, a MARL framework designed for this collective RNA design task. CoCoRNA uses a centralized training with decentralized execution strategy (Kraemer & Banerjee, 2016). It also introduces a novel search-augmented exploration mechanism to enhance learning efficiency. We also provide theoretical guarantees of convergence under specific conditions.
- 💡 We rigorously evaluate CoCoRNA on the `Rfam` dataset, along with extensive ablation studies. Under identical time constraints, CoCoRNA achieves a $97.75\%$ success rate, significantly surpassing the best baseline by $27.25$ percentage points ($38.7\%$ relative improvement). Even when the strongest baseline receives 20 times more computational time, CoCoRNA still maintains an $8.21$ percentage point lead ($9.2\%$ relative advantage). Importantly, CoCoRNA achieves an average solution time of only $3.24$ seconds, which is over 70 times faster than the best-performing baseline.

## 2 PRELIMINARIES

**RNA secondary structure representation**  In computational biology, RNA secondary structures are commonly represented using dot-bracket notation. Unpaired nucleotides appear as dots (`.`), while base pairs are indicated by matching parentheses. An opening parenthesis (`(`) marks the $5'$ end of a base pair, and a closing parenthesis (`)`) marks the $3'$ end. For example, `((...))` represents a hairpin loop, with paired bases at both ends and unpaired nucleotides in the middle.

**Decentralized partially observable Markov decision process**  We model the fully cooperative MARL problem as a decentralized partially observable Markov decision process (Oliehoek et al., 2008). Formally, this is defined by the tuple $\langle \mathcal{N}, \mathcal{S}, \mathcal{O}^i, \mathcal{A}^i, P, R, \gamma \rangle$, where:

- $\mathcal{N} = \{1, \ldots, N\}$: The set of $N$ agents.
- $\mathcal{S}$: The global state space.
- $\mathcal{O}^i$: Local observation space of agent $i$, with joint space $\mathcal{O} = \mathcal{O}^1 \times \cdots \times \mathcal{O}^N$.
- $\mathcal{A}^i$: The action space for agent $i$. The joint action space is $\mathcal{A} = \mathcal{A}^1 \times \cdots \times \mathcal{A}^N$.
- $P : \mathcal{S} \times \mathcal{A} \times \mathcal{S} \rightarrow [0, 1]$: The state transition probability.
- $R : \mathcal{S} \times \mathcal{A} \rightarrow [R_{\min}, R_{\max}]$: The shared reward function.
- $\gamma \in [0, 1]$: The discount factor.

At each time step $t$, the environment is in state $s_t \in \mathcal{S}$. Each agent $i \in \mathcal{N}$ observes a local state $o_t^i \in \mathcal{O}^i$. Based on its policy $\pi^i(a_t^i \mid o_t^i)$, each agent selects an action $a_t^i \in \mathcal{A}^i$. Together, these individual actions form the joint action $\boldsymbol{a}_t = (a_t^1, \ldots, a_t^N)$. The environment then transitions to a new state $s_{t+1}$ with probability $P(s_{t+1} \mid s_t, \boldsymbol{a}_t)$, and all agents receive a shared reward $R(s_t, \boldsymbol{a}_t)$. The goal is to find an optimal joint policy $\boldsymbol{\pi} = (\pi^1, \ldots, \pi^N)$ that maximizes the expected discounted return $G = \mathbb{E}\left[\sum_{t=0}^{\infty} \gamma^t R(s_t, \boldsymbol{a}_t)\right]$.

**Centralized training with decentralized execution (CTDE)**  This is a widely used paradigm in MARL (Kraemer & Banerjee, 2016). It combines the advantages of centralized and decentralized approaches. During training, agents have access to global information, which helps them evaluate and improve their local policies. However, at execution time, each agent selects actions based only on its local observations. CTDE enables agents to consider joint dynamics during training, addressing the non-stationarity issue common in multi-agent environments. It also promotes cooperative behavior. Decentralized execution further helps reduce dimensionality and improves scalability by minimizing the need for global information and communication.

**Actor-Critic methods with CTDE**  As the key components of the standard Actor-Critic framework (Barto et al., 1983), the Critic network estimates the value function, evaluating how good the

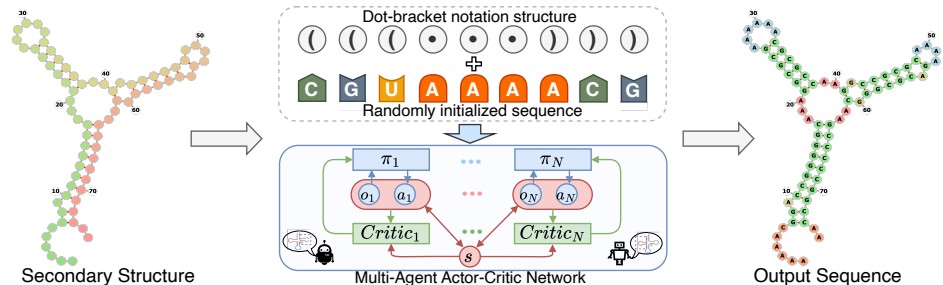

Figure 2: Overview of the CoCoRNA workflow.

current policy is; while the Actor learns and updates its policy using the Critic's value estimates. Under CTDE, this framework naturally extends to multi-agent systems, leading to centralized Critics and decentralized Actors (Lowe et al., 2017; Iqbal & Sha, 2019). During training, the centralized Critic leverages global information to provide accurate value estimates. These estimates guide the policy updates of each agent's decentralized Actor. At execution time, agents act independently, each relying solely on its local observations.

## 3 PROPOSED METHOD

The workflow of CoCoRNA is presented in Figure 2 and its pseudocode is provided in Appendix A. In the following paragraphs, we will delineate its key algorithmic components step by step.

### 3.1 PROBLEM DECOMPOSITION

In a canonical MDP, a policy $\pi : \mathcal{S} \to \mathcal{A}$ maps states to actions. Ideally, the state space $\mathcal{S}$ should capture all information relevant for decision-making. RNA design decisions depend primarily on two factors: the target RNA secondary structure, and the current RNA sequence. These factors matter because nucleotide positions have interdependencies due to base-pairing constraints. Thus, the state space $\mathcal{S}$ must combine RNA sequence and structural information:

$$\mathcal{S} = \{(s_{\texttt{seq}}, s_{\texttt{struct}}) \mid s_{\texttt{seq}} \in \{\texttt{A}, \texttt{U}, \texttt{G}, \texttt{C}\}^{\ell}, s_{\texttt{struct}} \in \{\,\textbf{.}\,, \textbf{(}, \textbf{)}\,\}^{\ell}\}, \tag{1}$$

where $s_{\texttt{seq}}$ is the nucleotide sequence, and $s_{\texttt{struct}}$ is the target secondary structure in dot-bracket notation. For sequences of length $\ell$, the total number of possible states is $|\mathcal{S}| = (4 \times 3)^{\ell}$, covering all possible sequence-structure combinations. Such exponential growth results in a high-dimensional policy space, complicating policy learning and exploration.

To disentangle this complexity, we propose using MARL within the CTDE framework. We decompose the RNA design problem into multiple subproblems, assigning them to different agents who solve these tasks cooperatively. Each agent then focuses only on specific positions or substructures. This targeted approach reduces the dimensionality of each agent's state and policy spaces. In this work, we propose two decomposition schemes:

- **Position-based decomposition**: The RNA sequence is split into $n$ segments, each assigned to a distinct agent. Each agent decides nucleotides exclusively for its segment.
- **Structure-type-based decomposition**: Each agent is assigned to a particular structural type (e.g., stems, loops). Agents design nucleotides solely at positions corresponding to their structural assignment.

Each agent receives a lower-dimensional local observation containing only a fragment of the RNA sequence and structure around its current position. Formally, an agent's observation space $\mathcal{O}$ is:

$$\mathcal{O} = \left\{ (o_{\texttt{seq}}, o_{\texttt{struct}}) \,\middle|\, o_{\texttt{seq}} \in (\{\texttt{A}, \texttt{U}, \texttt{G}, \texttt{C}, \emptyset\})^{m}, o_{\texttt{struct}} \in (\{\,\textbf{.}\,, \textbf{(}, \textbf{)}\,, \emptyset\})^{m} \right\}, \tag{2}$$

where $m$ is the length of the observation window (an odd integer). The symbol $\emptyset$ serves as a placeholder for positions beyond sequence boundaries. For agent $i$, its local observation $o_t^i$ is:

$$o_t^i = \left( s_{\texttt{seq}}^{[i-\kappa,\, i+\kappa]},\ s_{\texttt{struct}}^{[i-\kappa,\, i+\kappa]} \right), \tag{3}$$

where $\kappa = \frac{m-1}{2}$. $o_t^i$ consists of the local nucleotide and structural subsequences centered at position $i$. To handle variable-length inputs, sequences are padded with $\emptyset$ to a maximum length $l_{\max}$. This local observation provides sufficient context for the agent's decision.

> **Discussion:** By decomposing the complex RNA design task into smaller subproblems, we leverage the collective strengths of MARL. Each agent specializes either in a sequence segment or a specific structural type, enabling more focused and efficient policy learning.

### 3.2 POLICY OPTIMIZATION

We adopt multi-agent proximal policy optimization (MAPPO) (Yu et al., 2022) as our base algorithm due to its robust performance in various MARL tasks. Our framework provides each agent with two distinct networks: an Actor for decision-making, and a Critic for policy evaluation. Importantly, each agent maintains its own network parameters, enabling independent learning and specialized adaptation to its specific design role.

**Actor network** Each agent's Actor network receives a local observation $o_t^i$ as input, which provides relevant context around the agent's assigned position. The network then outputs a probability distribution over the discrete action space:

$$\mathcal{A} = \{\texttt{A}, \texttt{U}, \texttt{G}, \texttt{C}\}, \tag{4}$$

where each action corresponds to selecting a nucleotide type. We denote agent $i$'s policy as $\pi_{\theta^i}(a_t^i \mid o_t^i)$, where $\theta^i$ is the parameters of its Actor. During interaction with the environment, each agent samples an action $a_t^i$ from its Actor's distribution and executes it by placing the selected nucleotide.

**Critic network** The Critic network takes the global state $s_t$ as input, including both the current RNA sequence and its target secondary structure. We process these two components separately using two convolutional modules. Their outputs are concatenated and fed into subsequent Critic layers to produce a scalar value estimate $V(s_t)$. This estimate reflects the expected cumulative future rewards under the current joint policy $\boldsymbol{\pi}$:

$$V^{\boldsymbol{\pi}}(s_t) = \mathbb{E}_{\boldsymbol{\pi}}\left[\sum_{k=0}^{\infty}\gamma^k r_{t+k} \,\bigg|\, s_t\right]. \tag{5}$$

Using the Critic's value estimates, we calculate the temporal difference (TD) error $\delta_t = r_t + \gamma V(s_{t+1}) - V(s_t)$. Next, we apply generalized advantage estimation (GAE) (Schulman et al., 2015) to compute advantage estimates $A_t$:

$$A_t = \sum_{k=0}^{\infty}(\gamma\lambda)^k \delta_{t+k}, \tag{6}$$

where $\lambda \in [0, 1]$. These advantage estimates guide updates to each agent's Actor parameters $\theta^i$. Specifically, the MAPPO objective maximizes the following expression:

$$\max_{\theta^i} \mathbb{E}\left[\min\left(\frac{\pi_{\theta^i}(a_t^i \mid o_t^i)}{\pi_{\theta_{\mathrm{old}}^i}(a_t^i \mid o_t^i)}A_t, \mathrm{clip}\left(\frac{\pi_{\theta^i}(a_t^i \mid o_t^i)}{\pi_{\theta_{\mathrm{old}}^i}(a_t^i \mid o_t^i)}, 1-\epsilon, 1+\epsilon\right)A_t\right)\right], \tag{7}$$

where $\epsilon$ is a small positive constant controlling the PPO clipping. The convergence of our algorithm is established by the following lemma, assuming standard conditions for policy gradient convergence are satisfied (Konda & Tsitsiklis, 1999; Sutton et al., 1999). The detailed proof is in Appendix B.

**Lemma 3.1.** *Consider* COCORNA*'s multi-agent actor-critic algorithm, where a centralized Critic estimates the joint advantage function, and decentralized policies are updated according to the gradient:*

$$g = \mathbb{E}_{\boldsymbol{\pi}}\left[\sum_i \nabla_{\theta^i} \log \pi^i(a^i \mid o^i) A(s, \boldsymbol{a})\right]. \tag{8}$$

*Suppose the joint policy $\boldsymbol{\pi}$ is differentiable, uses an appropriate learning rate, and the Critic is compatible with $\boldsymbol{\pi}$ (Konda & Tsitsiklis, 1999; Sutton et al., 1999). Then, with probability 1, the algorithm converges to a local maximum of the expected joint return $J^{\boldsymbol{\pi}}$. That is:*

$$\liminf_{k\to\infty}\left\|\nabla_\theta J^{\boldsymbol{\pi}}(\theta_k)\right\| = 0 \quad \text{w.p. 1.} \tag{9}$$

**Discussion:** The centralized Critic uses global information to estimate values, guiding local policy updates of each agent. This strategy ensures individual actions align effectively with the overall team objective. By combining centralized value estimation and decentralized decision-making, we promote cooperative behavior despite agents acting independently based on local observations.

## 3.3 REWARD FUNCTION

We formulate RNA inverse design as a MDP which treats RNA design as a navigation task through the large design space. Our reward function is inspired by standard navigation problems, explicitly encouraging both exploration and optimization.

First, we use the ViennaRNA (Lorenz et al., 2011), a minimum free energy (MFE) folding algorithm, to predict the secondary structure $\mathbf{x}_f$ of the current RNA sequence. We then compute the Hamming distance, $H(\mathbf{x}_f, \mathbf{x}_t)$, between this predicted structure and the target structure $\mathbf{x}_t$ as: $H(\mathbf{x}_f, \mathbf{x}_t) = \sum_{i=1}^{\ell} \mathbb{1}(\mathbf{x}_f^i \neq \mathbf{x}_t^i)$, where $\mathbb{1}(\cdot)$ is an indicator function returning 1 if $\mathbf{x}_f^i \neq \mathbf{x}_t^i$, and 0 otherwise.

Reward function consists of two components: an intermediate reward and a final reward. At each step $t$, the multi-agent team selects actions, and the environment provides an intermediate reward reflecting the normalized reduction in Hamming distance from the previous time step. If the Hamming distance reaches 0, it means the predicted structure exactly matches the target structure. We then assign a substantial final reward $C$, and the episode terminates. Formally, the reward $R_t$ is defined as:

$$R_t = \begin{cases} (H_{t-1} - H_t)/\ell, & \text{if } H_t > 0, \\ C, & \text{if } H_t = 0, \end{cases} \tag{10}$$

where $C > 0$ is a constant. The intermediate reward encourages stepwise improvements, rewarding incremental reductions in structural differences. The substantial final reward $C$ strongly incentivizes agents to achieve a perfect match, thereby aiding them in overcoming local optima.

## 3.4 SEARCH-AUGMENTED EXPLORATION

The enormous RNA design space often causes cold-start issues for learning-based methods (Wang et al., 2024). In the early training stages, random policies often generate low-quality experiences, which limit learning efficiency. To address this, we introduce a heuristic called search-augmented exploration (SAE). SAE employs a limited greedy search early in training to enhance the quality of experiences stored in the replay buffer. Specifically, after an agent selects an action $a$ based on its current policy $\pi$ and receives reward $r$, we perform a brief local search. This search checks whether an alternative action $a'$ can yield a higher reward $r'$. If such an improved action is found, we replace the original experience tuple $(o, a, r)$ with $(o, a', r')$. By improving data quality early on, SAE accelerates the agents' learning progress.

In practice, COCORNA applies SAE during the first 30% of training, effectively mitigating cold-start challenges by exposing agents to more rewarding experiences earlier. Importantly, SAE does not introduce additional learning objectives or modify the original optimization goals. Therefore, SAE preserves MAPPO's convergence guarantees, as detailed in Section 3.2 and Appendix B.

## 4 EMPIRICAL STUDY

### 4.1 EXPERIMENTS ON RNA INVERSE DESIGN

**Dataset**   We evaluate COCORNA on RNA secondary structures from the `Rfam` dataset (Runge et al., 2024). We compute pairwise edit distances among $65,000$ structures and apply mini-batch $K$-means to partition them into 100 clusters. One cluster (650 structures) serves as the test set, while the remaining data are filtered to remove structures highly similar to the test set, yielding $60,000$ training structures. The minimum edit distance between training and test structures is 111, with an average of 188.42. To ensure robustness and prevent data leakage, we also construct six alternative splits by randomly selecting different clusters as test sets (details in Appendix F.1).

**Experimental setup**   We use position-based decomposition with $n = 4$ agents, where each RNA structure is divided into four subsequences of approximately equal length, each assigned to one agent.

Table 1: Evaluation of CoCoRNA and baseline methods on the RNA inverse design task. The time limit for CoCoRNA is set to 30 seconds. The suffixes in baseline names indicate different time constraints; for example, "−600s" denotes a time limit of 600 seconds per structure.

| Method | Number of solved samples | Mean success rate | Average diversity |
|---|---|---|---|
| CoCoRNA | **635.40 ± 2.58** | **97.75%** | **0.405** |
| LEARNA−30s | 63.80 ± 3.03 | 9.82% | 0.311 |
| LEARNA−60s | 132.80 ± 3.35 | 20.43% | 0.320 |
| LEARNA−300s | 368.20 ± 6.18 | 56.65% | 0.339 |
| LEARNA−600s | 382.20 ± 4.82 | 58.80% | 0.336 |
| Meta-LEARNA-Adapt−30s | 381.20 ± 5.63 | 58.65% | 0.377 |
| Meta-LEARNA-Adapt−60s | 451.40 ± 4.98 | 69.45% | 0.390 |
| Meta-LEARNA-Adapt−300s | 564.20 ± 4.09 | 86.80% | 0.392 |
| Meta-LEARNA-Adapt−600s | 582.00 ± 4.98 | 89.54% | 0.400 |
| SAMFEO−30s | 458.20 ± 6.10 | 70.50% | 0.361 |
| SAMFEO−60s | 488.60 ± 7.64 | 75.17% | 0.388 |
| SAMFEO−600s | 560.00 ± 2.76 | 86.15% | 0.394 |
| antaRNA−60s | 87.80 ± 2.70 | 13.51% | 0.361 |
| antaRNA−600s | 110.00 ± 6.67 | 16.92% | 0.390 |
| antaRNA−1200s | 171.40 ± 5.03 | 26.37% | 0.387 |
| OmniGenome−60s | 261.4 ± 4.04 | 40.22% | 0.389 |
| OmniGenome−600s | 502.20 ± 4.21 | 77.26% | 0.380 |
| OmniGenome−1200s | 532.40 ± 4.51 | 81.91% | 0.388 |

An episode has at most 400 steps and is considered successful if the agents collectively generate a sequence that folds into the target structure. We train the model for a total of 25 million steps. During evaluation, we generate sequences for each test structure with up to 15 attempts under a 30-second time limit. Network architectures and detailed hyperparameters are provided in Appendix D.

**Metrics & baselines** We adopt success rate and diversity as the primary evaluation metrics. A successful design is defined as an exact structural match to the target (i.e., Hamming distance = 0). To quantify the diversity of solutions, we compute the average pairwise normalized Hamming distance among the generated sequences. We compare CoCoRNA with five state-of-the-art RNA inverse design algorithms: antaRNA(Kleinkauf et al., 2015), SAMFEO(Zhou et al., 2023), LEARNA and Meta-LEARNA-Adapt(Runge et al., 2019), and OmniGenome(Yang et al., 2025). OmniGenome is a recent large-scale foundation model with 186M parameters, functioning as a generative model assisting genetic algorithms for RNA inverse design. antaRNA and SAMFEO are stochastic optimization methods, both requiring significant optimization time. LEARNA and its meta-learning variant, Meta-LEARNA-Adapt, perform online learning per sequence, demanding instance-specific policy training and additional computational resources. To ensure fair efficiency comparisons, we evaluate these baselines under multiple time constraints (30s, 60s, 300s, 600s, 1200s). Note that we give CoCoRNA the most strict time limit 30 seconds in all experiments.

**CoCoRNA schieves the highest success rate** Table 1 reports the overall performance, averaged across five independent runs. Under a strict 30-second time limit, CoCoRNA attains a success rate of 97.75%, representing a substantial improvement of 27.25 absolute percentage points (38.7% relative) over the strongest baseline at the same budget (SAMFEO−30s). Even when extending baseline budgets to 600 seconds—20 times longer than CoCoRNA's time limit—CoCoRNA still surpasses the strongest competitor (Meta-LEARNA-Adapt−600s) by 8.21 absolute points (9.2% relative). These results establish CoCoRNA as the most reliable method for RNA inverse design.

**CoCoRNA is highly efficient** Beyond accuracy, CoCoRNA also demonstrates remarkable efficiency. The average solution time is only 3.24 seconds per successful design, which is over 70 times faster than Meta-LEARNA-Adapt (averaging 234.2 seconds at the 600-second limit). Figure 3 further visualizes the time distribution and its correlation with sequence length, showing that CoCoRNA consistently solves targets faster than all baselines. Interestingly, despite being a large-scale generative model with 186M parameters, OmniGenome does not outperform classical stochastic optimization methods, highlighting the difficulty of accurately capturing the sequence–structure mapping. Moreover, the traditional optimizer antaRNA performs poorly (success rate 26.37% at 1,200s), often converging to local optima on long, complex structures.

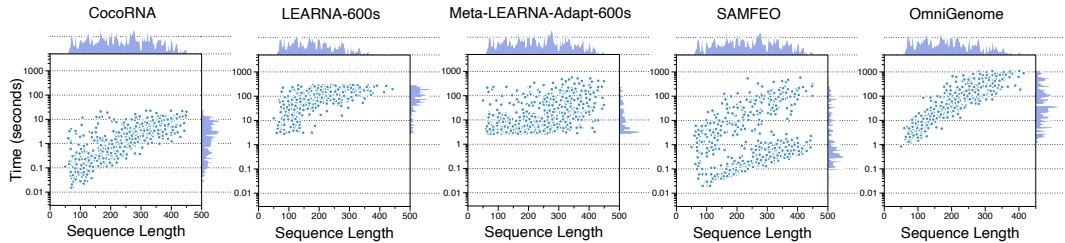

Figure 3: CPU time distribution of CoCoRNA and four baselines over successfully solved cases.

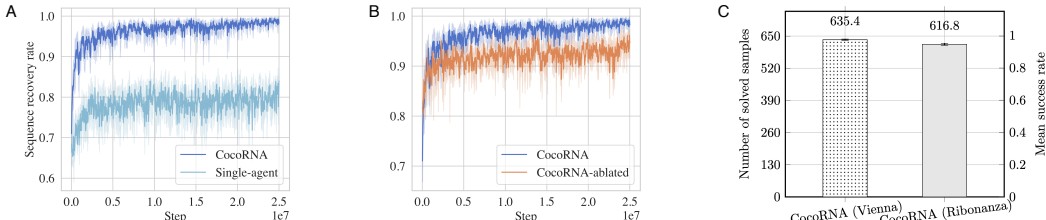

Figure 4: **A**. Comparison of sequence recovery rates during training between the `Single-agent` variant and CoCoRNA. **B**. Effect of removing the SAE heuristic from CoCoRNA. **C**. Cross-oracle validation experiment using RibonanzaNet.

**CoCoRNA maintains good diversity**    In addition to high success and efficiency, CoCoRNA produces diverse solutions. As summarized in Table 1, CoCoRNA achieves the highest average pairwise normalized Hamming distance ($0.405$), surpassing all baselines. This indicates that CoCoRNA not only finds correct solutions efficiently but also explores a broader sequence space, which is desirable for downstream applications requiring diverse candidate designs.

**Other biological sequence design tasks**    In addition, we apply CoCoRNA to solve other biological sequence design problems in the widely-used Design-Bench benchmark (Trabucco et al., 2022), including TF Bind 8, TF Bind 10 (Barrera et al., 2016), ChEMBL (Gaulton et al., 2012), and UTR (Sample et al., 2019). Experimental results demonstrated that CoCoRNA have shown promising results as well. This gives us sufficient confidence about the generalization capability of CoCoRNA. Due to page limit, details and results are provided in Appendix E.

### 4.2 ABLATION STUDIES

We perform ablation studies to dissect the contributions of different design choices in CoCoRNA.

**Multi-agent architecture yields clear gains**    To assess the benefit of collective design, we compare CoCoRNA with a `Single-agent` variant that uses a single RL agent to control all sequence positions. As shown in Figure 4 (left panel), the `Single-agent` variant consistently underperforms CoCoRNA across training, achieving substantially lower sequence recovery rates (SRR). This highlights the effectiveness of decomposing the task among multiple agents, which facilitates exploration and accelerates convergence.

**SAE stabilizes and accelerates training**    We further evaluate a variant of CoCoRNA without the SAE heuristic (`CoCoRNA-ablated`). The learning curves in Figure 4 (right panel) show that this ablated model converges more slowly and exhibits higher variance. This indicates that SAE provides high-quality experiences early in training, significantly improving stability and efficiency. More detailed SAE-related analyses are deferred to Appendix F.3.

**CoCoRNA generalizes across different oracles**    We train CoCoRNA with ViennaRNA and evaluate it using RibonanzaNet (He et al., 2024), a state-of-the-art deep learning–based structure predictor trained on large-scale experimental data. This setting provides a stringent test of generalizability, since RibonanzaNet adopts a methodology fundamentally different from ViennaRNA's physics-based MFE calculations. As shown in Figure 4, CoCoRNA achieves a $94.90\%$ success rate under RibonanzaNet evaluation. The slight drop relative to ViennaRNA is expected, but the

consistently high performance indicates that COCORNA does not merely overfit to a single oracle. Instead, it captures generalizable principles of RNA sequence–structure relationships.

**Additional robustness analyses**   To comprehensively evaluate COCORNA's robustness, we also investigated alternative decomposition schemes, reward functions, numbers of agents, parameter-sharing strategies, dataset splits, and hyperparameter settings. Results consistently confirm the robustness and versatility of our framework. Due to space constraints, full details and discussions are presented in Appendix F.

## 5   RELATED WORKS

The RNA inverse design problem was first introduced by Hofacker et al. (1994b) in the Vienna RNA package, which uses an adaptive random walk algorithm. There are various follow-up approaches. For example, RNA-SSD (Andronescu et al., 2004b) enhanced sequence initialization and reduced complexity by dividing sequences into smaller substructures. INFO-RNA (Busch & Backofen, 2006b) further combined dynamic programming for sequence initialization with stochastic local search. Later approaches explored larger search spaces and incorporated additional constraints using evolutionary algorithms (Taneda, 2012; Esmaili-Taheri et al., 2014; Esmaili-Taheri & Ganjtabesh, 2015), Monte Carlo tree search (Yang et al., 2017; Portela, 2018), and simulated annealing (Matthies et al., 2012). Other notable methods utilized constraint programming (Garcia-Martin et al., 2013; 2015), ant colony optimization (Kleinkauf et al., 2015), or ensemble optimization (Zhou et al., 2023). However, these methods primarily rely on stochastic optimization guided by handcrafted heuristics. Such iterative search approaches are computationally expensive, with complexity rapidly increasing as sequence length grows. Moreover, partially due to the *no free lunch* theorem (Wolpert & Macready, 1997), these heuristics hardly generalize to new problems.

In contrast, machine learning methods can automatically discover effective heuristics from data. Recent learning-based methods have gained prominence, particularly generative models (Ingraham et al., 2019), large language models (Lin et al., 2023; Subramanian et al., 2024), and RL approaches (Angermueller et al., 2019). These methods are successfully applied in various biological design tasks, such as protein function optimization (Lutz et al., 2023; Wang et al., 2023), protein sequence design (Dauparas et al., 2022; Gao et al., 2024), and drug repurposing (Huang et al., 2024). For RNA secondary structure design, Eastman et al. (2018) first utilized RL with pre-trained policies, effectively replacing handcrafted heuristics. Later, Runge et al. (2019) combined RL with neural architecture search to simultaneously optimize network architectures and hyperparameters. They further refined generative policies through instance-specific fine-tuning. Recent generative methods like RNAinformer (Patil et al., 2024) show promising results but are computationally intensive, limiting practical applicability to sequences under 100 nucleotides and lacking open-source availability. Similarly, OmniGenome (Yang et al., 2025), a large-scale 186M-parameter generative model, tackles RNA structure design by learning direct structure-to-sequence mappings.

We clarify key distinctions between COCORNA and related methods. COCORNA employs a MARL framework, decomposing the RNA design problem into manageable sub-tasks solved cooperatively. While RNA-SSD (Andronescu et al., 2004b) also divides sequences into independently optimized subsequences, it does not effectively incorporate global structural information and thus often fails to achieve globally optimal solutions by combining local solutions. In contrast, COCORNA uses a centralized Critic with complete global context, guiding agents to jointly reach optimal solutions. Another related method, GameOpt (Bal et al., 2023), frames optimization as a competitive game among variables. Although both COCORNA and GameOpt adopt decomposition strategies, their approaches differ fundamentally: COCORNA trains dynamic policies for cooperative adaptation, whereas GameOpt searches for a fixed solution over predefined optimization variables.

## 6   CONCLUSION

We propose COCORNA, a collective design method that employs cooperative MARL to address the challenges of RNA secondary structure design. By formulating the RNA design task as a collective problem and decomposing it into multiple sub-tasks assigned to individual agents, COCORNA reduces the complexity faced by each agent, enhances policy exploration, and improves learning efficiency. Experimental results fully demonstrate the effectiveness and generalizability of COCORNA.

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

## A  PSEUDOCODE FOR COCORNA

Algorithm 1 presents the pseudocode for the proposed COCORNA method. COCORNA leverages the MAPPO algorithm, an extension of the PPO method tailored for multi-agent environments. In our implementation of COCORNA, each agent is equipped with an Actor network responsible for selecting actions based on local observations, while the centralized Critic network evaluates the global state to provide value estimates for policy updates.

---

**Algorithm 1** COCORNA: Cooperative Multi-Agent RNA Design

---

1: **Input:** Target RNA secondary structure dataset $\mathbb{X}$, maximum episodes $E$, maximum steps per episode $T$, horizon $H$, number of agents $n$
2: **Initialize** Actor networks $\pi_{\theta^i}$ and Critic network $V_\phi$ for each agent $i$
3: **Initialize** experience replay buffer $\mathcal{D}$
4: **for** episode $= 1$ to $E$ **do**
5:    Reset environment, randomly select a target RNA secondary structure $x_t$ from $\mathbb{X}$, and initialize RNA sequence $s_{\text{seq}}$
6:    Decompose the RNA design task among $n$ agents according to the chosen decomposition scheme
7:    **for** step $t = 1$ to $T$ **do**
8:      **for** each agent $i = 1$ to $n$ **in parallel do**
9:        Observe local observation $o_t^i$
10:       Select action $a_t^i \sim \pi_{\theta^i}(a_t^i \mid o_t^i)$
11:     **end for**
12:     Execute joint action $\boldsymbol{a}_t = (a_t^1, a_t^2, \ldots, a_t^n)$
13:     Update RNA sequence $s_{\text{seq}}$ with actions $\boldsymbol{a}_t$
14:     Predict folded structure $x_f$ using MFE folding algorithm
15:     Calculate Hamming distance $H_t = H(x_f, x_t)$
16:     Calculate reward $r_t$ based on $H_t$ and $H_{t-1}$
17:     Store transition $(\boldsymbol{o}_t, \boldsymbol{a}_t, r_t, \boldsymbol{o}_{t+1})$ in buffer $\mathcal{D}$
18:     **if** using SAE **then**
19:       Perform local search to find better actions $\boldsymbol{a}_t'$
20:       Update reward $r_t'$ based on improved Hamming distance $H_t'$
21:       Replace transition with $(\boldsymbol{o}_t, \boldsymbol{a}_t', r_t', \boldsymbol{o}_{t+1})$ in buffer $\mathcal{D}$
22:     **end if**
23:     **if** $H_t = 0$ **then**
24:       Assign final reward $C$ to all agents
25:       **Break**
26:     **end if**
27:     **if** buffer size reaches horizon $H$ **then**
28:       Update Actor networks $\pi_{\theta^i}$ and Critic network $V_\phi$ using mini-batch updates from buffer $\mathcal{D}$ with MAPPO
29:       Clear experience buffer $\mathcal{D}$
30:     **end if**
31:   **end for**
32: **end for**

---

## B  THEORETICAL ANALYSIS

### B.1  CONVERGENCE ANALYSIS

In this section, we provide a convergence analysis of the proposed algorithm. We demonstrate that, under certain conditions, the multi-agent actor-critic method used in COCORNA converges to a local maximum of the expected joint return.

In COCORNA, we employ a centralized Critic network to evaluate the joint policy. Specifically, we use the joint advantage function $A(s, \boldsymbol{a})$, defined as:

$$A(s, \boldsymbol{a}) = Q(s, \boldsymbol{a}) - V(s), \tag{11}$$

where $Q(s, \boldsymbol{a})$ denotes the joint action-value function and $V(s)$ is the state-value function. The joint policy gradient is then given by:

$$g = \mathbb{E}_{\boldsymbol{\pi}} \left[ \sum_i \nabla_\theta \log \pi^i(a^i \mid o^i) \, A(s, \boldsymbol{a}) \right] \tag{12}$$

$$= \mathbb{E}_{\boldsymbol{\pi}} \left[ \sum_i \nabla_\theta \log \pi^i(a^i \mid o^i) \, Q(s, \boldsymbol{a}) \right] - \mathbb{E}_{\boldsymbol{\pi}} \left[ \sum_i \nabla_\theta \log \pi^i(a^i \mid o^i) \, V(s) \right], \tag{13}$$

where $\boldsymbol{\pi}$ denotes the joint policy, and $\theta$ represents the parameters of the Actor networks. The second term of (13) can be expanded as:

$$g_V = -\mathbb{E}_{\boldsymbol{\pi}} \left[ \sum_i \nabla_\theta \log \pi^i(a^i \mid o^i) \, V(s) \right] \tag{14}$$

$$= -\sum_s d^{\boldsymbol{\pi}}(s) \sum_i \sum_{\boldsymbol{a}^{-i}} \boldsymbol{\pi}(\boldsymbol{a}^{-i} \mid o^{-i}) \sum_{a^i} \nabla_\theta \pi^i(a^i \mid o^i) \, V(s), \tag{15}$$

where $d^{\boldsymbol{\pi}}(s)$ denotes the discounted ergodic state distribution (Sutton et al., 1999), $\boldsymbol{a}^{-i}$ and $o^{-i}$ represent the joint actions and observations of all agents except agent $i$.

Due to the normalization property of the policy:

$$\sum_{a^i} \pi^i(a^i \mid o^i) = 1, \tag{16}$$

we have:

$$\sum_{a^i} \nabla_\theta \pi^i(a^i \mid o^i) = \nabla_\theta \sum_{a^i} \pi^i(a^i \mid o^i) = \nabla_\theta 1 = 0. \tag{17}$$

Therefore,

$$g_V = -\sum_s d^{\boldsymbol{\pi}}(s) \sum_i \sum_{\boldsymbol{a}^{-i}} \boldsymbol{\pi}(\boldsymbol{a}^{-i} \mid o^{-i}) \, V(s) \, \nabla_\theta 1 \tag{18}$$

$$= 0. \tag{19}$$

Thus, the policy gradient in (13) reduces to only the first term:

$$g = \mathbb{E}_{\boldsymbol{\pi}} \left[ \sum_i \nabla_\theta \log \pi^i(a^i \mid o^i) \, Q(s, \boldsymbol{a}) \right]. \tag{20}$$

Since the joint policy can be expressed as the product of individual agent policies:

$$\boldsymbol{\pi}(\boldsymbol{a} \mid s) = \prod_i \pi^i(a^i \mid o^i), \tag{21}$$

and using the logarithmic identity $\log \prod_i x_i = \sum_i \log x_i$, we can rewrite the policy gradient as:

$$g = \mathbb{E}_{\boldsymbol{\pi}} \left[ \sum_i \nabla_\theta \log \pi^i(a^i \mid o^i) \, Q(s, \boldsymbol{a}) \right] \tag{22}$$

$$= \mathbb{E}_{\boldsymbol{\pi}} \left[ \nabla_\theta \log \prod_i \pi^i(a^i \mid o^i) \, Q(s, \boldsymbol{a}) \right] \tag{23}$$

$$= \mathbb{E}_{\boldsymbol{\pi}} \left[ \nabla_\theta \log \boldsymbol{\pi}(\boldsymbol{a} \mid s) \, Q(s, \boldsymbol{a}) \right]. \tag{24}$$

This result shows that (24) is equivalent to the single-agent actor-critic policy gradient, where the multiple agents are considered as a single joint agent (or super-agent). Konda & Tsitsiklis (1999) proved the convergence of policy gradient algorithms under the condition that the policy $\boldsymbol{\pi}$ is differentiable and the algorithm parameters satisfy certain conditions.

Therefore, by following the gradient in (12), the multi-agent actor-critic algorithm converges to a local maximum of the expected joint return $J^{\boldsymbol{\pi}}$.

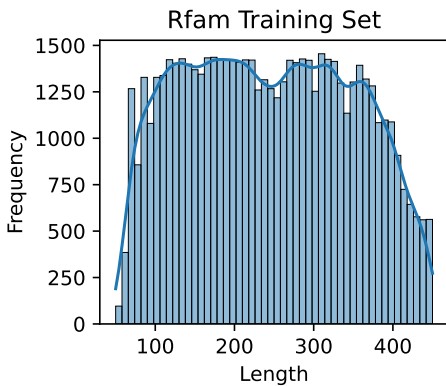 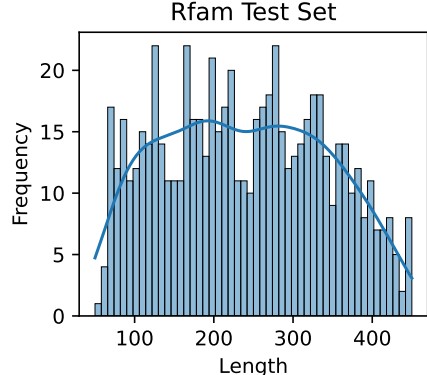

Figure 5: Length distributions of RNA secondary structures in different datasets.

### B.2 LIMITATION AND DISCUSSION

In theory, gradient-based deep learning and reinforcement learning methods generally guarantee convergence to local optima rather than global optima due to the non-convex nature of the optimization landscape. However, in practice, the ability of reinforcement learning algorithms to go beyond local optima and approach global optima depends significantly on how well they balance exploration and exploitation.

Specifically, CocoRNA is built upon PPO, which includes an entropy bonus in the objective function, which explicitly encourages the policy to maintain high entropy. This prevents the policy from becoming too deterministic too quickly, avoiding premature convergence to suboptimal policies and promoting continued exploration.

In the context of the RNA design task, the multi-agent framework of CocoRNA further aids in overcoming local optima. Unlike single-agent methods that mutate one nucleotide at one step, our method allows multiple agents to act simultaneously on different parts of the RNA sequence. This introduces greater diversity and increases the exploration of the state space, thereby increasing the opportunities to discover better policies.

## C DATASETS DETAILS

We use the Rfam dataset (Runge et al., 2024) to train and evaluate our algorithm. The dataset contains only RNA secondary structure information without explicit sequence labels, as the task is to design sequences that fold into the provided structures.

The Rfam dataset is constructed by applying RNA folding algorithms to RNA sequences from the Rfam database (Kalvari et al., 2021). The structures in this dataset are computationally predicted using tools like the ViennaRNA package (Lorenz et al., 2011), which produces secondary structures based on minimum free energy (MFE) folding. We randomly sample 60,000 RNA secondary structures as the training set and an additional 650 structures as the test set.

Figure 5 shows the length distributions of RNA secondary structures in the Rfam training set and test set. Table 2 provides statistical information about the RNA structures in these datasets, including the sample size, average length, maximum length, and minimum length of the RNA secondary structures.

## D ALGORITHM DETAILS AND HYPERPARAMETERS

This section provides a detailed description of the COCORNA algorithm, including network architectures, training details, and hyperparameter settings.

Table 2: Statistics of the datasets.

| Dataset | Sample Size | Average Length | Max Length | Min Length |
|---|---|---|---|---|
| Rfam (Training Set) | 60000 | 243.73 | 450 | 50 |
| Rfam (Test Set) | 650 | 238.78 | 450 | 50 |

Table 3: Training hyperparameters.

| Hyperparameter | Value |
|---|---|
| Learning Rate | 0.00001 |
| Discount Factor ($\gamma$) | 0.99 |
| GAE Parameter ($\lambda$) | 0.95 |
| PPO Clip Range ($\epsilon$) | 0.2 |
| Entropy Coefficient | 0.02 |
| Horizon Length | 256 |
| Batch Size | 512 |
| Number of Epochs | 10 |
| Gradient Clipping | 0.5 |

### D.1 NETWORK ARCHITECTURES

Each agent in the COCORNA framework is equipped with its own Actor and Critic networks. Although the Critic network leverages shared global information, the network parameters are not shared across agents. The Actor network takes local observations as input and outputs a discrete action (the nucleotide to be placed in the RNA sequence). The Critic network, on the other hand, processes the global state, which includes the full RNA sequence and secondary structure, and outputs a scalar value representing the estimated cumulative future reward for the entire system.

- **Actor Network:** The Actor network consists of several fully connected layers followed by a softmax output layer. Each agent's observation is passed through the network to compute a probability distribution over the action space $\{A, U, G, C\}$.

- **Critic Network:** The Critic network takes the entire RNA sequence and secondary structure as input. This input is processed through two separate CNN modules: one for the sequence and one for the structure. The outputs of these modules are concatenated and passed through a fully connected network to produce a value estimate for the entire system.

### D.2 TRAINING AND OPTIMIZATION

The training procedure follows on-policy reinforcement learning paradigm: experiences are collected over a fixed horizon, stored in a replay buffer, and subsequently used to update the Actor and Critic networks. We adopt PPO to optimize the policy by maximizing the clipped surrogate objective, which stabilizes training, and employ Generalized Advantage Estimation (GAE) to compute advantages with reduced variance.

Training on a single NVIDIA 4090 GPU requires approximately 8 to 10 hours. As shown in Appendix F.4, this can be reduced to 3 to 4 hours under a delayed reward formulation with negligible performance degradation. Inference is highly efficient, averaging only 3.24 seconds per successful design.

Table 4: Network hyperparameters.

| Hyperparameter | Value |
|---|---|
| Actor Network Layers | $[128, 64]$ |
| Critic Network Layers | $[256, 64]$ |
| CNN Filter Size | $[8, 4, 3]$ |
| CNN Number of Filters | $[32, 64, 64]$ |
| Activation Function | ReLU |
| Optimizer | Adam |

Table 5: Additional hyperparameters.

| Parameter | Value |
|---|---|
| Maximum Steps per Episode (Training) | 350 |
| Maximum Steps per Episode (Testing) | 400 |
| Observation Radius | 30 |
| Local Observation Window Size | 61 |
| Global State Dimension | 900 |
| Total Training Steps | $2.5 \times 10^7$ |

## D.3 HYPERPARAMETER SETTINGS

Table 3 and Table 4 summarize the key hyperparameters used in training the COCORNA model, including training parameters and network parameters.

## D.4 OTHER SETTINGS

Table 5 presents additional parameter settings used in the COCORNA framework. During training, the maximum number of steps per episode is set to 350, while for testing, it is increased to 400. The training is conducted for a total of 25 million steps. Each agent has an observation radius of 30, resulting in a local observation window size of 61. Since the maximum RNA structure length in the dataset is 450, we standardize the global state dimension to 900 (450 for the sequence information and 450 for the structure information).

We use the ViennaRNA package (Lorenz et al., 2011) version 2.6.4 to compute the folded RNA structures during the design process.

Table 6: Performance of COCORNA, Meta-LEARNA-Adapt-600s, and SAMFEO-600s across six different train-test splits. The mean and standard deviation are reported.

| Method | Number of solved samples | Mean success rate |
|---|---|---|
| COCORNA | $633.33 \pm 5.47$ | $97.44\%$ |
| Meta-LEARNA-Adapt-600s | $578.17 \pm 9.60$ | $88.95\%$ |
| SAMFEO-600s | $568.16 \pm 6.09$ | $87.41\%$ |

Table 7: Performance of CoCoRNA with different problem decomposition schemes on Rfam dataset across six different train-test splits. The mean and standard deviation over are reported.

| Method | Number of solved samples | Mean success rate |
|---|---|---|
| CoCoRNA-PBD | $633.33 \pm 5.47$ | 97.44% |
| CoCoRNA-SBD | $628.50 \pm 4.99$ | 96.69% |
| CoCoRNA-random | $596.33 \pm 6.70$ | 91.74% |

Table 8: Performance of CoCoRNA and ablated versions on Rfam dataset. The mean and standard deviation over five independent runs are reported.

| Method | Number of solved samples | Mean success rate |
|---|---|---|
| CoCoRNA | $635.40 \pm 2.58$ | 97.75% |
| CoCoRNA-ablated | $599.60 \pm 7.64$ | 92.25% |
| Single-agent version | $340.80 \pm 5.63$ | 52.43% |

## E  OTHER BIOLOGICAL SEQUENCE DESIGN TASKS

Beyond RNA inverse design, we further apply CoCoRNA to four biological sequence design tasks. These tasks are formulated as RL environments based on DesignBench (Trabucco et al., 2022). Specifically, they include the DNA functional optimization tasks TF Bind 8, TF Bind 10 (Barrera et al., 2016), and UTR (Sample et al., 2019), as well as the drug molecule design task ChEMBL (Gaulton et al., 2012). Apart from minor adjustments to the state space, action space, and hyperparameters such as the learning rate, CoCoRNA requires no additional modifications.

**Results**  Given that the algorithms in Section 4.1 are specifically tailored for RNA design, we can hardly adapt them to other tasks without nearly redesigning a new algorithm from scratch. Therefore, we compare CoCoRNA with its single-agent ablated variant to further validate the effectiveness of the multi-agent framework. The results, presented in Table 9, clearly show that the MARL framework of CoCoRNA consistently outperforms its single-agent counterpart across all tasks. Moreover, CoCoRNA successfully reaches the global optimum in TF Bind 8 and TF Bind 10. Considering that CoCoRNA was not originally designed for these tasks, these results further demonstrate its adaptability and effectiveness across diverse problem settings.

## F  ADDITIONAL RESULTS

In this section, we present a series of experiments under various settings to comprehensively demonstrate the effectiveness of CoCoRNA.

### F.1  DATASET SPLITTING

In the main experiments, we cluster the dataset based on structural similarity and randomly select one cluster out of 100 as the test set. To more rigorously rule out potential data leakage, we further construct six different train-test splits by randomly selecting six distinct clusters as test sets. Independent training and evaluation are then conducted on each split. Additionally, we evaluate two of the strongest baselines—Meta-LEARNA-Adapt and SAMFEO—on these test sets. The results are reported in Table 6.

The results show that CoCoRNA achieves an average success rate of 97.44% across these new train-test splits, which is highly consistent with the results reported in the main paper. These findings further demonstrate the strong generalization ability of CoCoRNA.

Table 9: Evaluation of CoCoRNA and the single-agent ablated version on biological sequence design tasks. Results represent the average of 8 independent runs, with standard deviations also reported.

|  | TF Bind 8 | TF Bind 10 | ChEMBL | UTR |
|---|---|---|---|---|
| CoCoRNA | $1.000 \pm 0.000$ | $1.000 \pm 0.000$ | $0.811 \pm 0.001$ | $0.810 \pm 0.001$ |
| Single-agent version | $0.901 \pm 0.093$ | $0.725 \pm 0.140$ | $0.595 \pm 0.166$ | $0.690 \pm 0.010$ |

## F.2 Using Different Decomposition Schemes

Table 7 provides the testing results for different problem decomposition schemes used in CoCoRNA. We employ three decomposition methods: Position-based decomposition (CoCoRNA-PBD), Structure-type-based decomposition (CoCoRNA-SBD) and random decomposition(CoCoRNA-random). For Position-based decomposition, the complete RNA structure is divided into approximately equal segments, with each segment assigned to a different agent. For Structure-type-based decomposition, agents are assigned specific structural types based on the dot-bracket notation; for example, if Agent 1 is responsible for non-paired structures, it sequentially designs nucleotides at all non-paired positions within the structure.

In our experiments, CoCoRNA-PBD and CoCoRNA-SBD only show minor performance differences, demonstrating the flexibility of our method concerning decomposition choices. For CoCoRNA-SBD, structure-type-based decomposition allows agents to focus more on specific structural elements, but the relative positional relationships between different agents may become completely disrupted as the design process progresses, potentially causing slight adverse effects on the learning process. Although CoCoRNA-SBD performed slightly worse than CoCoRNA-PBD in testing, both methods significantly outperformed the single-agent approach. CoCoRNA-random is worse than the other two decomposition methods. We hypothesize that this is because randomly assigning design positions is entirely unstructured. This increases the difficulty for agents to cooperate and interferes with the learning process. However, the performance of random decomposition does not completely collapse, suggesting that the MARL mechanism itself is robust.

## F.3 Ablation Study on SAE

Table 8 presents the testing results of two ablated versions of CoCoRNA. Removing the SAE method results in a slight performance decrease in CoCoRNA-ablated, demonstrating the effectiveness of the SAE approach. Additionally, as shown in the right plot of Figure 4, CoCoRNA-ablated continues to improve towards the end of the training process, suggesting that further training may yield better performance. The single-agent version shows significantly poorer performance. In fact, the single-agent version is similar to the LEARNA method but lacks architectural and hyperparameter optimizations and does not undergo additional training during the design phase.

Table 10: Performance of CoCoRNA and CoCoRNA-ablated under different reward settings on the Rfam dataset.

| Reward Setting / Method | Number of solved samples | Success rate |
|---|---|---|
| Delayed Reward / CoCoRNA | 634/650 | 97.54% |
| Delayed Reward / CoCoRNA-ablated | 621/650 | 95.54% |
| Terminal Reward / CoCoRNA | 604/650 | 92.92% |
| Terminal Reward / CoCoRNA-ablated | 570/650 | 87.69% |

## F.4 Ablation Study on Reward Signals

In addition to the standard reward signal, we test two different delayed/sparse reward settings on the Rfam dataset.

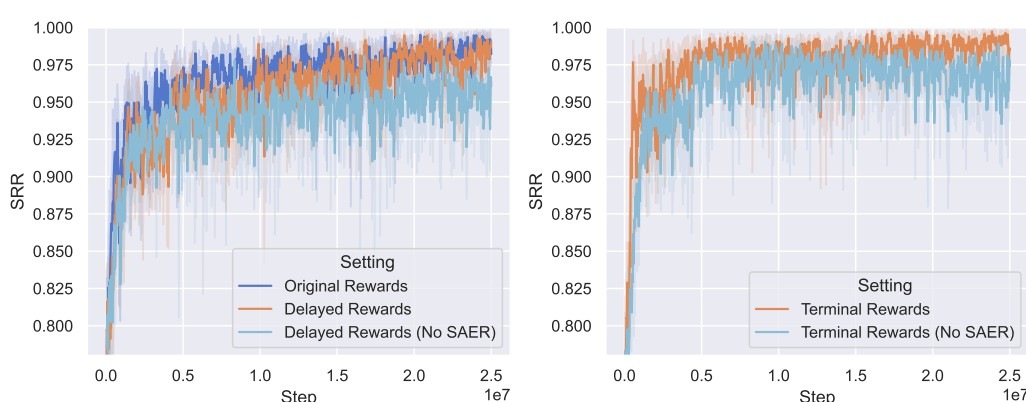

Figure 6: Learning curves under different reward settings on the Rfam dataset. Each experiment is performed over 6 independent runs with different random seeds. The shaded areas represent the 95% confidence intervals. **Left**: Learning curves using delayed reward (25). **Right**: Learning curves using terminal reward (26).

- **Delayed Reward:** The reward signal is calculated every 10 steps instead of every step. The original reward function (10) is modified as follows:

$$R_t = \begin{cases} \frac{H_{t-10}-H_t}{l}, & \text{if } H_t > 0 \text{ and } t \bmod 10 = 0, \\ C, & \text{if } H_t = 0 \text{ and } t \bmod 10 = 0, \\ 0, & \text{otherwise.} \end{cases} \quad (25)$$

- **Terminal Reward Only:** The reward is given only at the end of an episode or upon successful design. The reward function is defined as:

$$R_t = \begin{cases} 1 - \frac{H_t}{l}, & \text{if } H_t \neq 0 \text{ and } t \text{ reaches maximum steps (episode ends),} \\ C, & \text{if } H_t = 0, \\ 0, & \text{otherwise.} \end{cases} \quad (26)$$

We also test the impact of removing the SAE method under these two reward settings. The results are shown in Figure 6 and Table 10. We denote the ablated version without SAE as COCORNA-ablated. It can be observed that under the delayed reward setting, COCORNA experiences almost no performance loss. When only terminal rewards are available, the algorithm's performance decreases slightly due to the excessively sparse reward signal significantly increasing the difficulty of policy learning. Under both reward settings, COCORNA-ablated performs worse than COCORNA, demonstrating the effectiveness of the SAE method. Overall, this experiment shows that COCORNA is robust to different reward signals and performs well even under harsh reward settings. Notably, the delayed reward setting reduces the number of calls to the RNA folding algorithm to one-tenth of the original, significantly speeding up training.

Table 11: Performance of COCORNA under different agent size settings on the Rfam dataset.

| Number of Agents ($n$) | Number of solved samples | Success rate |
|:---:|:---:|:---:|
| $n = 2$ | 601/650 | 92.46% |
| $n = 4$ | 636/650 | 97.85% |
| $n = 6$ | 629/650 | 96.77% |
| $n = 8$ | 615/650 | 94.62% |

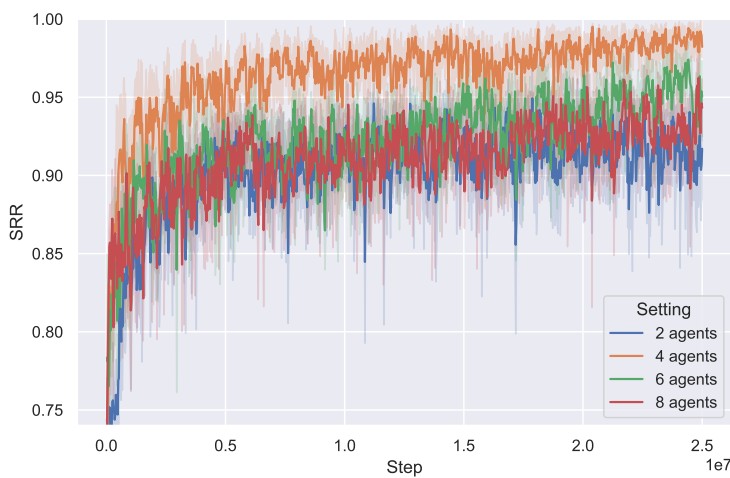

Figure 7: Learning curves of COCORNA under different agent size settings on the Rfam dataset. Each experiment is performed over 6 independent runs with different random seeds. The shaded areas represent the $95\%$ confidence intervals.

### F.5 EFFECT OF VARYING THE NUMBER OF AGENTS

In the main results presented earlier, we set the number of agents to $n = 4$. In this section, we adjust the parameter $n$ to examine the impact of different numbers of agents on the algorithm's performance. We experiment with four different settings: using 2, 4, 6, and 8 agents. Table 11 shows the performance of COCORNA under these different agent sizes on the Rfam dataset. The corresponding learning curves are presented in Figure 7.

From the results, we observe that using 4 or 6 agents achieves the best performance. When the number of agents is too small or too large, the performance tends to decrease. With too few agents, the benefits of multi-agent problem decomposition cannot be fully exploited, as each agent handles a larger portion of the task, leading to higher complexity per agent and potentially less efficient learning. On the other hand, with too many agents, the environment becomes more non-stationary from the perspective of each agent due to the increased interactions and dependencies among agents, making cooperation more challenging.

Table 12: Performance of COCORNA with independent and shared policies on the Rfam dataset.

| Method | Number of solved samples | Success rate |
|---|---|---|
| COCORNA-PBD | 636/650 | 97.85% |
| COCORNA-PBD-shared | 622/650 | 95.69% |
| COCORNA-SBD | 629/650 | 96.77% |
| COCORNA-SBD-shared | 621/650 | 95.54% |

### F.6 EFFECT OF SHARING POLICY PARAMETERS

We investigate the performance of COCORNA when agents share policy parameters under both decomposition methods. In these experiments, although different agents receive different observations, all agents share the same set of policy parameters. This approach is referred to as COCORNA-PBD-shared and COCORNA-SBD-shared for position-based decomposition and structure-type-based decomposition, respectively.

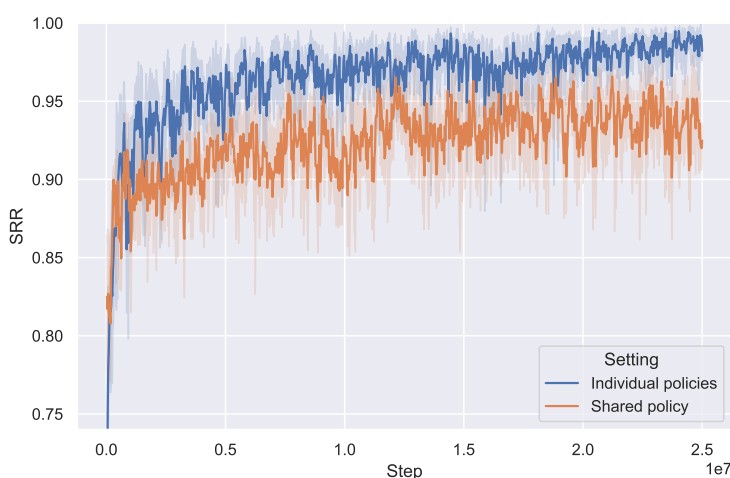

Figure 8: Learning curves of COCORNA-PBD and COCORNA-PBD-shared on the Rfam dataset. Each experiment is performed over 6 independent runs with different random seeds. The shaded areas represent the 95% confidence intervals.

The learning curves and test results are shown in Figure 8 and Table 12, respectively. From the results, we observe that enforcing shared policy parameters among multiple agents leads to a slight decrease in algorithm performance compared to using independent policies, although the gap is relatively small.

These results potentially suggest that different agents learn distinct local policies that are specialized for their assigned sub-tasks. When agents share policy parameters, the policy learning process is constrained because all agents must use the same policy function, despite facing different sub-tasks or local environments. This can limit the agents' ability to adapt their policies to their specific roles in the RNA design task. In contrast, when using independent policies, each agent can tailor its policy parameters to its particular sub-task without interference from the data of other agents, providing more flexibility in policy learning.

### F.7 TRAINING PARAMETERS

We observe that the algorithm is relatively sensitive to the learning rate compared to other hyper-parameters. In our main experiments, we use a learning rate of $1 \times 10^{-5}$, which is lower than the settings commonly used in RL algorithms. Figure 9 shows the learning curves when using higher learning rates. It can be seen that excessively high learning rates lead to unstable training and poor convergence.

In multi-agent environments, a high learning rate can exacerbate the non-stationarity of the environment. This issue arises because each agent's policy update affects the environment dynamics perceived by other agents. With higher learning rates, these changes become more abrupt, making it difficult for agents to adapt and learn stable policies. Consequently, the training process becomes highly unstable.

For other important hyperparameters, such as the discount factor $\gamma$, the GAE parameter $\lambda$, and the PPO clip range $\epsilon$, we use standard values commonly adopted in the literature. These settings are detailed in Appendix D.

### F.8 OTHER RESULTS

Figure 10 shows how the number of solved samples increases as a function of the allotted runtime. The median solving time of CocoRNA is only $1.47$ s.

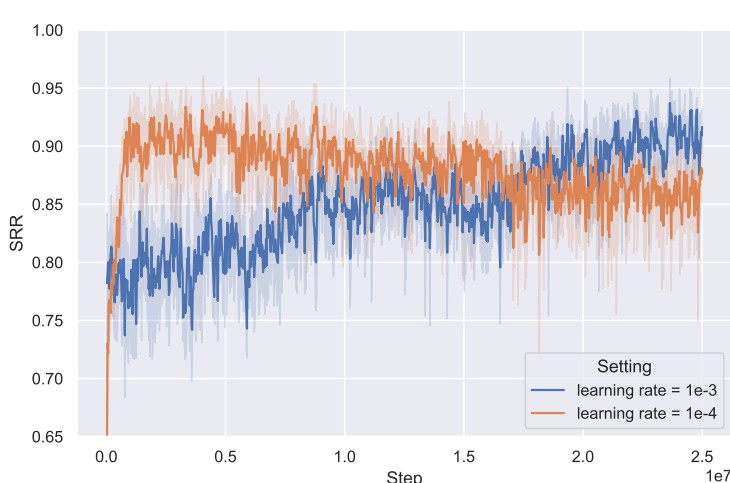

Figure 9: Learning curves of COCORNA using different learning rates. Each experiment is performed over 6 independent runs with different random seeds. The shaded areas represent the $95\%$ confidence intervals.

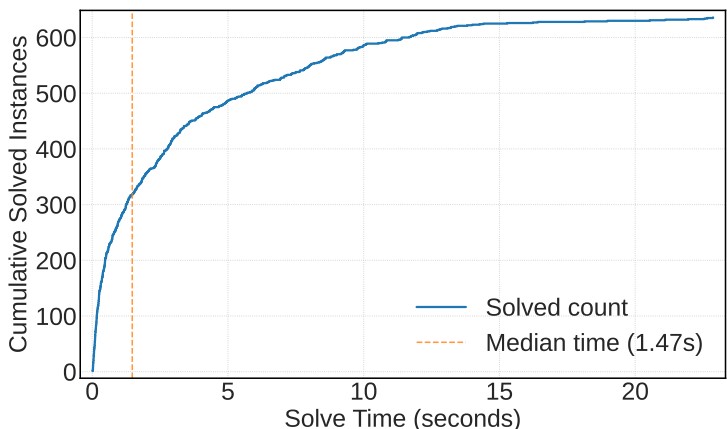

Figure 10: Number of solved samples as a function of the allotted runtime.

Figure 11 shows the distribution of solving times for COCORNA, LEARNA, and Meta-LEARNA-Adapt on the Rfam dataset, including only the sequences that were successfully solved.

Figure 12 shows the number of redesign attempts (iterations) $n_i$ required to design different RNA structures on the Rfam dataset using COCORNA, LEARNA-600s, and Meta-LEARNA-Adapt-600s. Table 13 presents the corresponding statistical information. For the majority of sequences (95.54%), our method succeeds within a single episode. In contrast, the two baseline methods require numerous iterative optimizations. Although Meta-LEARNA-Adapt performs meta-learning on the entire training set, only 3.38% of sequences can be solved in a single attempt.

Table 13: Distribution of RNA sequences by redesign attempts $n_i$ intervals for different methods on the Rfam dataset.

| Method | $n_i = 1$ | $1 < n_i \leq 15$ | $n_i > 15$ |
|---|---|---|---|
| CoCoRNA | 621 | 29 | 0 |
| LEARNA | 1 | 16 | 633 |
| Meta-LEARNA | 22 | 137 | 498 |

## G  DISCUSSION

### G.1  GRAPH NEURAL NETWORKS

Many biological molecules are well-suited to be modeled as graphs due to their inherent structural properties. Transforming RNA structures into graph representations and utilizing Graph Neural Networks (GNNs) could potentially better capture the relationships between different nucleotides, enabling more effective policy learning. Graph-based representations naturally model the interactions and dependencies within RNA structures, which could enhance the agent's decision-making process.

However, employing more complex network architectures like GNNs also introduces higher computational and memory overheads, potentially reducing training efficiency. In our work, although we did not use GNNs, we addressed the issue of partial observations by employing CNNs within the centralized Critic to extract global structural information. This approach helps mitigate the limitations of local observations by providing a holistic view of the RNA structure, thereby supporting coordinated policy optimization without incurring excessive computational costs. Nevertheless, integrating GNNs with reinforcement learning-based methodologies represents a promising direction for future research.

### G.2  LIMITATIONS

While CoCoRNA demonstrates promising results in RNA secondary structure design, there are several limitations.

First, it is non-trivial to design an accurate and reliable reward model given the complexity of biological systems. We may resort to large-scale pretrained models, e.g., RNA/protein language models. However, these models are often too large and thus are computationally expensive. One potential solution is to go with a model-based RL, where an explicit model of the environment is learned and used to predict future states and rewards more efficiently.

Another potential limitation is the decomposition method considered in the current CocoRNA. Given the high-dimensional nucleotide design space, it will be more promising to study an adaptive decomposition mechanism. This might be achieved by designing hierarchical policies, where high-level agents make decisions about task decomposition and low-level agents focus on specific sub-tasks.

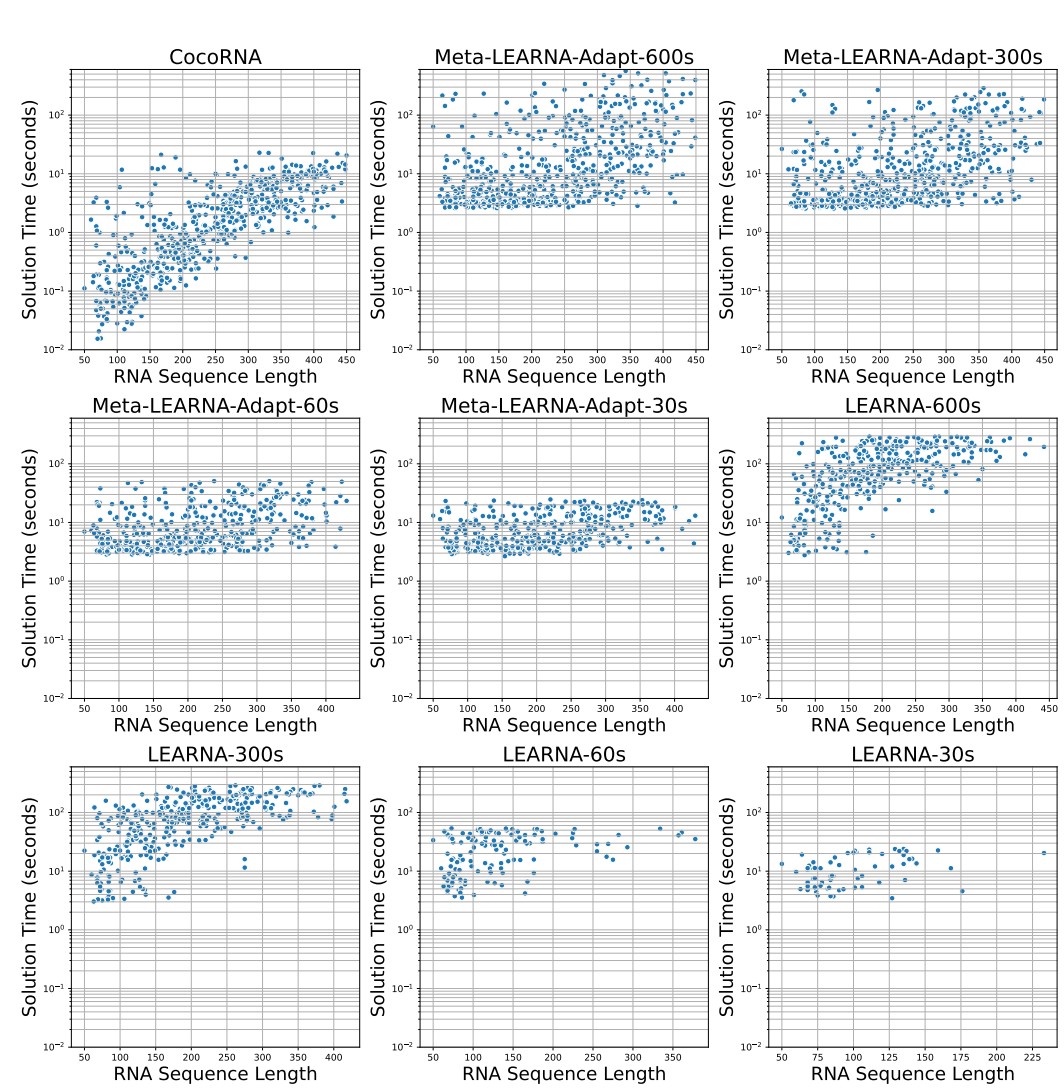

Figure 11: Distribution of solving times for different RNA design methods on the Rfam dataset.

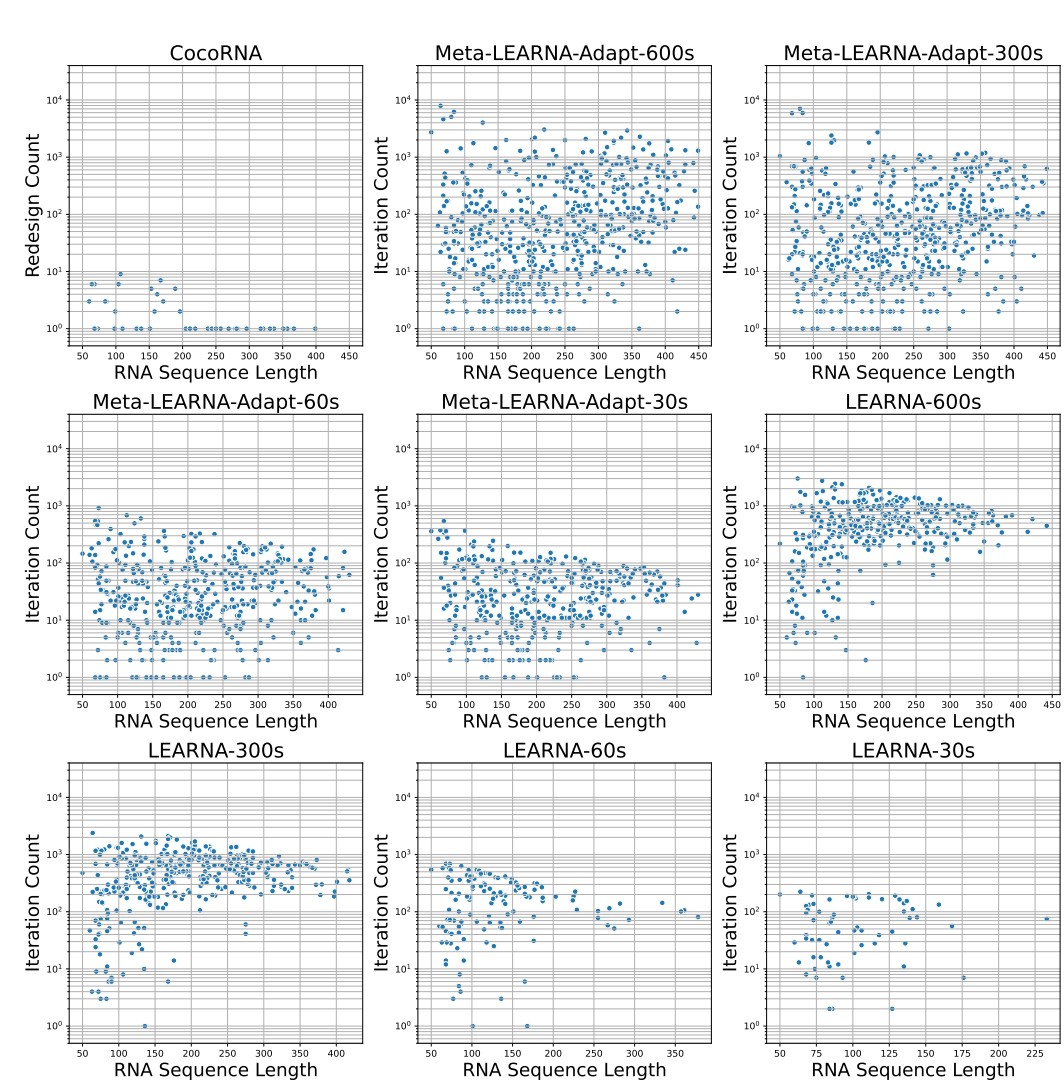

Figure 12: Distribution of redesign attempts (iterations) required by different RNA design methods on the Rfam dataset.

