# OpenReview forum: "CocoRNA: Collective RNA Design with Cooperative Multi-agent Reinforcement Learning"
_ICLR.cc/2026/Conference — Submitted to ICLR 2026_

### Official Review · Reviewer_PR76 · 2025-10-29

**Soundness:** 3
**Presentation:** 3
**Contribution:** 2
**Rating:** 4
**Confidence:** 4

**Summary:**

The paper proposes CocoRNA, a multi-agent cooperative framework for secondary structure based RNA design. The authors subdivide an RNA structure in dot-bracket format into several sub-tasks, each solved by an individual agent, while using a global critic network to ensure that the design considers the overall structure during design. The method is evaluated on the Rfam dataset and compared to five different competitors. On this benchmark CocoRNA outperforms all other approaches even when using a much stricter time limit. Furthermore, CocoRNA is also adapted and evaluated on other design tasks provided with the Design-Bench benchmark with promising results. In an ablation study, the authors confirm the gains of a multi-agent system compared to a single agent as well as their newly introduced SAE strategy for early training.

**Strengths:**

- Compared to existing RL approaches for secondary structure based RNA design, the multi-agent approach appears interesting and makes sense.
- The reported performance on the Rfam dataset is strong.
- The decomposition of the problem into sub-tasks while keeping a global measure of performance is a good idea.

**Weaknesses:**

1. I’m a bit concerned about the timeliness of the approach. Specifically, using the dot-bracket notation is a clear limitation and excludes design for pseudoknots, or other pairing schemes.
2. When using the data from [1], why do the authors not evaluate against [1]? In the paper, it seems that libLEARNA is superior compared to Meta-LEARNA-Adapt on the task of RNA design for nested structures. Also RNAPond [2] could be an interesting competitor here. I think there are also further competitors that could be evaluated, e.g. [3,4] but it might be too much to run all these in the limited amount of time during rebuttal.
3. One motivation in the introduction is about delayed rewards and misleading auxiliaries. However, the proposed reward formulation seems to match the formulation used in [5] quite exactly. Or am I missing something?
4. While the authors show strong results on the Rfam dataset, the structural diversity might be rather low. A good additional benchmark could be the Eterna100 (version 2) [6] in this case. While not the best benchmark for generative approaches, I would recommend evaluations on it here, since it specifically covers corner cases of the thermodynamic model of RNAfold, which could provide further insights into CocoRNA’s performance.
5. The authors provide visual examples on an external website. Looking at the designs, it seems that most sequences rarely contain ‘A’ nucleotides. At least there is a clear bias towards G, C, and surprisingly U. The authors should at least mention this somewhere. But I think this could also be a general limitation of the method, since the authors do not include any measure to avoid bad sequences (GC biases can strongly influence function). In the best case, a desired GC-content of the designed sequences would be included (which was e.g. done in [1] in a very simple way that seemed to work well already).
6. In this regard, the diversity measure might also be a bit misleading since it is only based on the Hamming Distance between the designed sequences.
7. It would also be interesting to see evaluations based on families. Using RNAfold, it is not too computationally expensive to even fold all sequences in the Rfam database and provide a family based evaluation scheme.

[1] Runge, F., Franke, J., Fertmann, D., Backofen, R., & Hutter, F. (2024). Partial RNA design. Bioinformatics, 40(Supplement_1), i437-i445.

[2] Yao, H. T., Waldispühl, J., Ponty, Y., & Will, S. (2021, April). Taming disruptive base pairs to reconcile positive and negative structural design of RNA. In RECOMB 2021-25th international conference on research in computational molecular biology.

[3] Minuesa, G., Alsina, C., Garcia-Martin, J. A., Oliveros, J. C., & Dotu, I. (2021). MoiRNAiFold: a novel tool for complex in silico RNA design. Nucleic acids research, 49(9), 4934-4943.

[4] Yang, X., Yoshizoe, K., Taneda, A., & Tsuda, K. (2017). RNA inverse folding using Monte Carlo tree search. BMC bioinformatics, 18(1), 468.

[5] Runge, F., Stoll, D., Falkner, S., & Hutter, F. Learning to Design RNA. In International Conference on Learning Representations.

[6] Koodli, R. V., Rudolfs, B., Romano, J., Wayment-Steele, H. K., Dunlap IV, W. A., Eterna Participants, & Das, R. (2021). Redesigning the EteRNA100 for the Vienna 2 folding engine. BioRxiv, 2021-08.

**Questions:**

1. It seems like the sequence is randomly initialized. Have the authors thought about / tested different initialization methods? SAMFEO for example, uses an initialization with GC pairs and A for unpaired nucleotides if I remember correctly which itself could already solve many tasks without further predictions when using RNAfold. I would guess that this would further improve performance of CocoRNA (although it doesn’t solve the problem with GC-content during design described above).
2. In the plots of Figure 3 (and similarly Figure 10). It seems that CocoRNA’s runtime scales quadratic with sequence length (although being much faster than the other methods). If I remember correctly, linear scaling with sequence length was one of the claims of the original LEARNA paper. Can the authors elaborate on that?
3. While already stated above, what do the authors think about including e.g. pseudoknots in the design? Using dot-bracket notation, this would blow-up the state-space I guess? What about using matrix representations as many recent DL folding engines output L x L matrices like done in the cited work of RNAinformer?
4. Is there any limitation on sequence length for the designs?

---

> ### Author Response · Authors · 2025-11-27
> **Response to reviewer PR76 (Part 1)**
>
> We sincerely thank the reviewer for the detailed and valuable comments. We greatly appreciate the recognition of the motivation, novelty, and soundness of our work.
>
> Below, we summarize all of the reviewer’s concerns and provide our responses accordingly.
>
> ---
>
> **W1 & Q3. Matrix representation and pseudoknots.**
>
> The reviewer correctly notes that the dot-bracket representation (used in our primary implementation) has limitations. However, the CocoRNA *framework* is representation-agnostic. It can readily adopt a matrix-based representation, which naturally handles pseudoknots and aligns with modern DL folding engines.
>
> **Matrix-based CocoRNA:** We implemented a matrix-based variant using a three-channel input (Structure, Sequence, Position) processed by a CNN, and retrained the model.
>
> |  | Number of solved samples  | Mean success rate |
> | --- | --- | --- |
> | CocoRNA(dot-bracket) | $635.40 \pm 2.58$ | $97.75\\%$ |
> | CocoRNA(matrix) | $630.2 \pm 4.67$ | $96.95\\%$ |
>
> The matrix variant maintains strong performance.
>
> **Pseudoknot design:** We further tested the matrix variant on the Kleinkauf et al. dataset (250 pseudoknotted structures), using IPknot [1] as the oracle. We fine-tuned CocoRNA using only $10\\%$ of the original training budget.
>
> |  | Results on test sets #1-3 | Mean success rate |
> | --- | --- | --- |
> | CocoRNA(pseudoknots) | $91, 94, 93$ | $92.67\\%$ |
> | antaRNA | $61, 65, 67$ | $64.33\\%$ |
>
> CocoRNA significantly outperforms antaRNA (the only baseline supporting pseudoknots) by **over 28 percentage points**. These results demonstrate the flexibility of the CocoRNA framework and its applicability to complex RNA structures.
>
> **W2 & W4. Comparison with additional baselines and Eterna100 datasets**
>
> Following the suggestions, we added libLEARNA and RNAPond as baselines (600s limit) and included the Eterna100 benchmark.
>
> |  | Mean success rate (Rfam) | Mean success rate (Eterna100) |
> | --- | --- | --- |
> | CocoRNA | $97.75\\%$ | $69.6\\%$ |
> | libLEARNA | $90.03\\%$ | $67.4\\%$ |
> | RNAPond | $43.91\\%$ | $25.8\\%$ |
>
> libLEARNA performs strongly but still lags behind CocoRNA. On Eterna100, the margin is smaller because this dataset specifically targets corner cases of the thermodynamic model (as noted by the reviewer). CocoRNA maintains state-of-the-art performance across both natural and synthetic benchmarks.
>
> **W5 & W6. Can CocoRNA control specific properties (e.g., GC content)?**
>
> Yes. The reviewer’s **W5** correctly observes a GC bias in the default model (Mean $\text{GC}=0.62$). This is common as G-C pairs are energetically favorable. CocoRNA can naturally incorporate additional objectives via reward shaping to control such properties. We demonstrate this by optimizing for GC content ($r_{\texttt{GC}}$, target 0.5) and Minimum Free Energy ($r_{\texttt{MFE}}$) via modified reward functions. These rewards are given only when the structural design is successful.
>
> - $r_{\texttt{GC}}=\alpha \lvert c_{\texttt{GC}} - c^{t} \rvert$, where $c_{\texttt{GC}}$ is the normalized GC content of the designed sequence, $c^{t} = 0.5$ is the target GC content, and $\alpha$ is the scaling factor.
> - $r_{\texttt{MFE}} = -\beta E$ , where $E$ is the minimum free energy of the sequence and $\beta$ is the scaling factor.
>
> |  | Mean success rate | Mean G-C content | Mean MFE |
> | --- | --- | --- | --- |
> | CocoRNA (structrue) | $97.75\\%$ | $0.62$ | $-125.1$ |
> | CocoRNA (structrue + G-C content) | $97.20\\%$ | $0.51$ | - |
> | CocoRNA (structrue + MFE) | $97.07\\%$ | - | $-158.5$ |
> | CocoRNA (structrue + G-C content + MFE) | $97.26\\%$ | $0.53$ | $-151.2$ |
>
> These results demonstrate that CocoRNA can effectively control GC content and optimize MFE without compromising structural accuracy, addressing the observed bias.
>
> ---
>
> > [1] Sato, K., Kato, Y., Hamada, M., Akutsu, T., & Asai, K. (2011). IPknot: fast and accurate prediction of RNA secondary structures with pseudoknots using integer programming. *Bioinformatics*, *27*(13), i85-i93.
> >

---

> ### Author Response · Authors · 2025-11-27
> **Response to reviewer PR76 (Part 2)**
>
> **W7. Family-Based Evaluation**
>
> We constructed four family-specific test sets (500 structures each, derived from Rfam seed, deduplicated).
>
> |  | Success rate |
> | --- | --- |
> | rRNA | $99.6\\%$ |
> | tRNA | $99.8\\%$ |
> | Ribozyme | $94.0\\%$ |
> | Riboswitch | $99.6\\%$ |
>
> CocoRNA achieves over $99\\%$ success on three families. The lower initial rate on Ribozymes ($94.0\\%$) confirms that design difficulty is family-dependent. When extending the time limit to 600s, the Ribozyme success rate recovers to $98.8\\%$. This indicates that certain Ribozyme structures present a more rugged optimization landscape, requiring extended search time for CocoRNA to locate the global optimum.
>
> **W3. How does CocoRNA address delayed rewards and misleading auxiliary signals? Is this achieved through reward engineering?**
>
> The reviewer is correct that our reward *function* is standard (similar to [5]). Our improvements do *not* stem from reward engineering, but rather from the fundamental changes to the learning dynamics introduced by the multi-agent architecture.
>
> 1. **Shorter effective horizons:** Long trajectories make credit assignment difficult. By dividing the task, each agent performs fewer steps before a full sequence is constructed and evaluated. This shortens the effective horizon, reducing the delay between action and reward, thereby alleviating the delayed-reward problem.
> 2. **Enhanced Exploration:** Our MARL framework enables parallel, collaborative exploration. Simultaneous modifications by multiple agents introduce greater diversity in the search, crucial for navigating the complex energy landscape.
>
> Our ablations (App F.4, p. 20) show CocoRNA is robust across different reward designs, confirming that the benefits arise from the collaborative architecture itself.
>
> **Q1. Can performance be improved through specialized initialization strategies?**
>
> We evaluated CocoRNA using the initialization strategy suggested by the reviewer, and the results are shown below. We did not observe any noticeable improvement. Our interpretation is that while certain initialization schemes may accelerate the design process for a subset of relatively easy structures, they do not help with structures that are inherently difficult to solve. As a result, the overall success rate remains largely unchanged.
>
> |  | Mean success rate |
> | --- | --- |
> | CocoRNA (original) | $97.75\\%$ |
> | CocoRNA (with initialization) | $97.84\\%$ |
>
> **Q2. How does the runtime scale with sequence length?**
>
> CocoRNA's runtime involves: (1) sequence evaluation (RNAfold, $\mathcal{O}(N^3)$) and (2) exploration (searching the space).
>
> Empirically (Fig 3), CocoRNA exhibits approximately quadratic scaling. This contrasts with some baselines which may appear closer to linear. This difference in observed scaling occurs because those baselines are significantly less efficient. They spend a much larger proportion of time searching, which dilutes the relative contribution of the $\mathcal{O}(N^3)$ folding cost. Because CocoRNA explores efficiently and finds solutions quickly, the time spent in RNAfold evaluations becomes a larger fraction of the *total* (but much shorter) runtime, leading to the observed quadratic trend. It is worth noting that CocoRNA remains significantly faster in absolute terms across all lengths.
>
> **Q4. Tests on long sequences (>500 nt).**
>
> CocoRNA is not inherently limited by sequence length. While our implementation uses a fixed input dimension (450nt), longer sequences can be handled via a sliding-window representation.
>
> To demonstrate scalability, we tested the *original* CocoRNA model (**without retraining**) on 500 RNA sequences from Rfam ranging from 500nt to 971nt, using a sliding-window approach (600s limit, 5 repetitions).
>
> |  | Results | Mean success rate |
> | --- | --- | --- |
> | CocoRNA (long RNA dataset) |  $457, 460, 462, 465, 467$ | $92.4\\%$ |
>
> Despite being tested on sequences significantly longer (up to $2\times$) than those seen during training, CocoRNA achieves a $92.4\\%$ success rate. This highlights the framework’s strong zero-shot generalization capabilities.

---

### Official Review · Reviewer_EEwV · 2025-10-31

**Soundness:** 3
**Presentation:** 3
**Contribution:** 2
**Rating:** 2
**Confidence:** 3

**Summary:**

This paper introduces COCORNA, a cooperative multi-agent reinforcement learning framework for RNA inverse folding, designing RNA sequences that fold into a given secondary structure. The key innovation is task decomposition. The RNA sequence is divided into segments or structural parts. These agents work together under a "CTDE" paradigm, using "MAPPO" for policy optimization and search-augmented exploration. COCORNA achieves a 97.75% success rate on the Rfam dataset, and it significantly outperforming state-of-the-art methods in both accuracy and speed. It also generalizes well to other biological sequence design tasks.

**Strengths:**

- Here utilizes the multi-agent reinforcement learning framework for RNA inverse folding, offering a fresh perspective on tackling complex biological sequence design problems.
- This work achieves an impressive 97.75% success rate on the Rfam dataset, significantly outperforming existing methods (e.g., LEARNA, SAMFEO, antaRNA) by large margins.
- Besides, provides theoretical convergence guarantees under standard assumptions, enhancing the credibility and understanding of the proposed method.

**Weaknesses:**

- Rewards signal depends on external tools like ViennaRNA. The rewards can become misleading, if the predicted structure is inaccurate. No uncertainty modeling or robustness to predictor error.
- Here only uses secondary structure as the design target, while 3D structure is more crucial for RNA function in reality. Ignoring it may lead to biologically non-functional designs.
- Theoretical analysis only guarantees local convergence, not global optimality. Like most RL methods, it can get stuck in suboptimal policies, especially in sparse reward setting.

**Questions:**

- Why did authors choose fixed position-based or structure-based decomposition? Or have explored adaptive or learnable decomposition strategies? Is there exist other decomposition schemes and their impact on the performance?
- Search-augmented exploration improves early training but involves local greedy search. Could this mislead the policy towards short-term improvements?

---

> ### Author Response · Authors · 2025-11-27
>
> We sincerely thank the reviewer for the valuable comments and for recognizing the novelty and significance of our work.
>
> We summarize all of the reviewer’s questions and concerns below and provide our responses.
>
> ---
>
> **W1. Is ViennaRNA an unreliable oracle?**
>
> We address the concern about oracle dependence in two ways:
>
> 1. **Empirical generalization:** We demonstrated strong cross-oracle generalization (Fig 4C). We evaluated sequences with *RibonanzaNet*, a SOTA deep learning model fundamentally different from ViennaRNA (physics-based MFE). The high success rate ($94.90\\%$) provides stringent evidence that CocoRNA is not merely overfitting to ViennaRNA's idiosyncrasies.
> 2. **Framework flexibility:** CocoRNA is oracle-agnostic and can easily incorporate additional objectives. We demonstrate this by optimizing for GC content ($r_{\texttt{GC}}$, target 0.5) and Minimum Free Energy ($r_{\texttt{MFE}}$) via modified reward functions. These rewards are given only when the structural design is successful.
>     - $r_{\texttt{GC}}=\alpha \lvert c_{\texttt{GC}} - c^{t} \rvert$, where $c_{\texttt{GC}}$ is the normalized GC content of the designed sequence, $c^{t} = 0.5$ is the target GC content, and $\alpha$ is the scaling factor.
>     - $r_{\texttt{MFE}} = -\beta E$ , where $E$ is the minimum free energy of the sequence and $\beta$ is the scaling factor.
>
> |  | Mean success rate | Mean G-C content | Mean MFE |
> | --- | --- | --- | --- |
> | CocoRNA (structrue) | $97.75\\%$ | $0.62$ | $-125.1$ |
> | CocoRNA (structrue + G-C content) | $97.20\\%$ | $0.51$ | - |
> | CocoRNA (structrue + MFE) | $97.07\\%$ | - | $-158.5$ |
> | CocoRNA (structrue + G-C content + MFE) | $97.26\\%$ | $0.53$ | $-151.2$ |
>
> These results confirm the framework is robust and adaptable to multi-objective optimization.
>
> ---
>
> **W2. “3D structure is more crucial for RNA function. Ignoring it may lead to biologically non-functional designs.”**
>
> We respectfully disagree with the premise. Secondary structure is central to RNA biology. Many functional properties are determined by 2D topology, which often dictates 3D conformation. In contrast, high-resolution 3D RNA data is extremely scarce, and 3D prediction tools are significantly less mature than their protein counterparts. Therefore, focusing on secondary structure is the current standard and biologically well-motivated frontier for RNA design. It is also worthwhile noting that, as shown in **W1**, CocoRNA is flexible and can incorporate 3D objectives as reliable oracles become available.
>
> **W3. How well CocoRNA enables finding global versus local optima.**
>
> CocoRNA’s ability to find global optima stems from two synergistic mechanisms: globally-aware training and enhanced exploration.
>
> 1. **Globally-coordinated learning (CTDE):** While agents have local views (decentralized execution), the centralized critic has **full observability** during training. It evaluates the *entire* sequence/structure and the quality of the *joint* action. This provides a globally informed gradient to each actor. The critic explicitly teaches agents to avoid locally optimal moves (e.g., forming an easy pair) if they preclude critical long-range interactions necessary for the global structure.
> 2. **Effective exploration (MARL):** The RNA energy landscape is rugged. Our single-agent ablation (Fig 4A) shows a monolithic agent struggles. By decomposing the problem, the multi-agent approach parallelizes exploration and introduces diverse search patterns, making it far more effective at escaping local optima.
> 3. **Empirical proof:** Our success metric is strict: success requires finding the *global* optimum (Hamming distance = 0) according to the oracle. The $97.75\\%$ success rate on a challenging held-out set (Table 1) empirically validates CocoRNA’s effectiveness in finding global optima.
>
> **Q2. Regarding search-augmented exploration(SAE).**
>
> We clarify that SAE is specifically designed to mitigate the cold-start problem. More importantly, SAE is applied *only* during the first $30\\%$ of training. This prevents the policy from overfitting to the greedy search heuristic and avoids convergence to local optima.
>
> Empirically, SAE enhances the ability to find the global optimum. Our success metric requires an exact match (global optimum). The ablation (Table 8) shows SAE improves the final success rate from $92.25\\%$ to $97.75\\%$, demonstrating that this early augmentation ultimately improves the quality of the converged policy.

---

### Official Review · Reviewer_DtVm · 2025-10-31

**Soundness:** 4
**Presentation:** 4
**Contribution:** 2
**Rating:** 8
**Confidence:** 4

**Summary:**

This work targets the inverse design of RNA secondary structures, an important biological sequence design task. The main contribution is the novel application of multi-agent RL to this task. The proposed approach, CocoRNA, achieves an impressive empirical performance.

**Strengths:**

* RNA design is an important problem to study and an interesting application of RL
* The main strength of the paper is the strong empirical performance in both number of solved sequences and time-to-solution
* Applying RL to RNA design has been explored before, but the application of multi-agent RL is novel and has several advantages. Decomposing the RNA design problem into sub-problems for different agents to solve to reduce state dimensionality and policy spaces is also an interesting idea.
* The authors identify specific shortcomings of current RL strategies and motivate the use of multi-agent RL and the decomposition into sub-tasks well.
* Both success rate and design speed are evaluated
* The ablation study for the multi-agent architecture and proposed exploration heuristic confirm the design choices behind CocoRNA.
* Related methods are covered well and key differences discussed
* Written with a very clear language and structure
* The illustrative Figures 1 & 2 are quite helpful
* The authors cover limitations in Appendix G2. Moving this discussion into the main paper would be good.

**Weaknesses:**

* Previous work evaluated also on the Eterna100 dataset, why is this omitted here?
* Code to reproduce the experiments or even an algorithm implementation is not provided. Some hyperparameter settings and details on architecture and optimization is provided though.
* Novelty in terms of methodology and problem setting is limited. Due to the strong performance, I still think this work is interesting for the ICLR community to be aware of which methods work well in applications.

**Questions:**

* "As demonstrated by our experiments in Section 4, existing approaches typically require hours of computation", where do approaches require hours of computation?
* Does the Rfam dataset you use differ from the one used in previous works?
* How were the hyperparameter settings for CocoRNA chosen?
* Table 1 compares CocoRNA with a 30s time limit (the most strict one) to baselines at 30s time limits and above. A plot that shows the #solved samples across different time limits would improve the evaluation and likely underline the strong performance and quick solution time (as seen in Figure 2) of CocoRNA.

---

> ### Author Response · Authors · 2025-11-27
>
> We sincerely thank the reviewer for the detailed and valuable comments. We greatly appreciate the positive assessment and the recognition of the motivation, novelty, and significance of our work.
>
> We summarize all of the reviewer’s questions and concerns below and provide our responses accordingly.
>
> ---
>
> **W1. Evaluation using Eterna100**
>
> We initially prioritized benchmarks based on real biological data, as Eterna100 is a synthetic benchmark specifically engineered to stress-test thermodynamic models.
>
> Following the reviewer’s suggestion, we evaluated CocoRNA on Eterna100 and included libLEARNA, a stronger baseline.
>
> |  | Mean success rate (Eterna100) |
> | --- | --- |
> | CocoRNA | $69.6\\%$ |
> | LEARNA | $49.8\\%$ |
> | RNAPond | $25.8\\%$ |
> | Meta-LEARNA-Adapt | $61.6\\%$ |
> | libLEARNA | $67.4\\%$ |
>
> CocoRNA achieves the best performance. The margin over libLEARNA is smaller than on Rfam, which is expected given Eterna100‘s focus on extreme, human-designed corner cases. Nonetheless, these results demonstrate CocoRNA’s robustness even on these highly challenging problems.
>
> **W2. Reproducibility**
>
> We respectfully clarify that our code was included in the initial submission’s supplementary materials.
>
> **W3. Novelty**
>
> We appreciate the opportunity to clarify our novelty. Our contributions are twofold: establishing a new paradigm for biological design using MARL, and developing the specific technical innovations required to realize this paradigm effectively.
>
> 1. **A new paradigm: cooperative MARL for biological design.** CocoRNA is the *first* framework to formulate RNA inverse design as a cooperative MARL problem. This is a significant conceptual contribution. It addresses the intractable search space of RNA design by transforming it into a distributed, collaborative optimization problem. It opens a new research direction for computational biology.
> 2. **Specific methodological innovations.** Applying MARL here was non-trivial and required domain-specific innovations beyond off-the-shelf algorithms:
>     - **Novel Dec-POMDP formulation:** We developed a new state/action representation tailored for collective RNA design (Sec 3.1).
>     - **Structured decomposition:** We introduced position- and structure-based decomposition schemes. Our ablations (App F.2) show these structured approaches are critical, far outperforming naive random decomposition.
>     - **Principled curriculum learning (SAE):** We designed SAE to address the severe ‘cold-start’ problem inherent in this complex domain.
>
> In summary, our novelty lies in the entire conceptual framework and the tailored system that enables MARL to establish a new state-of-the-art in RNA design.
>
> **Q1. Which methods require hours of computation?**
>
> We apologize for the inaccurate phrasing. While some prior methods (such as antaRNA and LEARNA) allow solving times of up to 1 hour or even 24 hours, we did not run experiments with such long time limits. We will correct this phrasing in the revised manuscript to avoid misunderstanding.
>
> **Q2. Does the Rfam dataset you use differ from the one used in previous works?**
>
> All our data are sourced from the Rfam dataset. However, we apply a **much stricter dataset splitting strategy** than prior works. Specifically, we construct **five distinct train–test splits**, each based on clustering RNA structures by similarity, to ensure robustness and to prevent any potential data leakage between training and test sets.
>
> **Q3. How were the hyperparameters chosen?**
>
> Most of our hyperparameters follow standard configurations commonly used in reinforcement learning. Some parameters were manually tuned with light heuristic adjustments, such as the learning rate and network size. We acknowledge that the chosen hyperparameters may not be fully optimal. However, the strong and stable performance across all experiments suggests that our framework is robust to these choices.
>
> **Q4. Additional suggestions regarding presentation**
>
> We thank the reviewer for the helpful suggestion. We have added a figure illustrating the **number of solved samples across different time limits**, and we have temporarily included it as **Figure 10** in **Appendix F.8** (page 24).
>
> If the paper is accepted (and we are granted one additional page), we will move this figure into the main text. Likewise, we will also relocate the discussion of limitations from **Appendix G** into the main manuscript.

---

### Official Review · Reviewer_Aczb · 2025-11-02

**Soundness:** 3
**Presentation:** 3
**Contribution:** 3
**Rating:** 4
**Confidence:** 2

**Summary:**

The paper proposes COCORNA, a multi-agent reinforcement learning framework for RNA inverse design. In their proposed method, the task is split into smaller sub-problems handled by multiple agents. A centralized critic coordinates learning, and a search-augmented exploration (SAE) improves early training. The method achieves a 97.75% success rate on Rfam, 70× faster than strong baselines. It also generalizes to other biological sequence design benchmarks.

**Strengths:**

They proposed the first multi-agent framework for RNA reverse design with solid theoretical foundation and relevant proof.

They demonstrated significant improvements in success rate and speed, supporting their claim with evidence.

Comprehensive ablation study and experimental details are presented and well-organized.

**Weaknesses:**

Missing comparison with new foundation models (e.g., RNAinformer).

Lacking deeper analysis of unsuccessful design cases.

Limited tests on long sequences (>500 nt).

**Questions:**

Could you involve more comparison to recent foundation models for RNA reverse design?

For the ~2.25% of structures that COCORNA fails to design within the time limit, could you provide a deeper analysis? What characteristics do these challenging structures share (e.g., length, complexity, specific structural motifs)?

While the cross-oracle validation with RibonanzaNet (94.90% success rate) is impressive, the 2.85 percentage point drop suggests some overfitting to ViennaRNA's MFE predictions. Could you elaborate on: (a) which types of structures show the largest performance gaps between oracles, and (b) potential strategies to make the approach more oracle-agnostic?

---

> ### Author Response · Authors · 2025-11-27
>
> We sincerely thank the reviewer for their valuable comments and for recognizing the novelty and significance of our work.
>
> We summarize the reviewer’s concerns below and provide our responses accordingly.
>
> ---
>
> **W1. Comparison with new foundation models**
>
> We have added a comparison with OmniGenome, a recently proposed RNA foundation model (Results in Table 1). The results are striking: CocoRNA (30s time limit) significantly outperforms OmniGenome (600s time limit), achieving a success rate that is over 20 percentage points higher. This demonstrates superior performance while operating under a time limit that is $20\times$ faster.
>
> Regarding RNAinformer, a direct comparison remains infeasible as the model and training code are not publicly available.
>
> **W2 & Q2. Do the unsuccessful design cases share any common characteristics?**
>
> We analyzed unsuccessful cases but found no obvious correlation with sequence length or simple structural motifs. Even short structures can be difficult. We hypothesize that these failures often involve structures inherently challenging for the underlying thermodynamic oracle. If only a minute fraction of sequences fold into the target structure according to the oracle, the design task becomes exceedingly difficult.
>
> We further investigated correlations between success rate and RNA family. We constructed four family-specific test sets (500 structures each, derived from Rfam seed, deduplicated).
>
> |  | Success rate |
> | --- | --- |
> | rRNA | $99.6\\%$ |
> | tRNA | $99.8\\%$ |
> | Ribozyme | $94.0\\%$ |
> | Riboswitch | $99.6\\%$ |
>
> CocoRNA achieves over $99\\%$ success on three families. The lower initial rate on Ribozymes ($94.0\\%$) confirms that design difficulty is family-dependent. When extending the time limit to 600s, the Ribozyme success rate recovers to $98.8\\%$. This indicates that certain Ribozyme structures present a more rugged optimization landscape, requiring extended search time for CocoRNA to locate the global optimum.
>
> **W3. Limited tests on long sequences (>500 nt).**
>
> CocoRNA is not inherently limited by sequence length. While our implementation uses a fixed input dimension (450nt), longer sequences can be handled via a sliding-window representation.
>
> To demonstrate scalability, we tested the *original* CocoRNA model (**without retraining**) on 500 RNA sequences from Rfam ranging from 500nt to 971nt, using a sliding-window approach (600s limit, 5 repetitions).
>
> |  | Results | Mean success rate |
> | --- | --- | --- |
> | CocoRNA (long RNA dataset) |  $457, 460, 462, 465, 467$ | $92.4\\%$ |
>
> Despite being tested on sequences significantly longer (up to $2\times$) than those seen during training, CocoRNA achieves a $92.4\%$ success rate. This highlights the framework’s strong zero-shot generalization capabilities.
>
> **Q3. On cross-oracle validation**
>
> The reviewer correctly notes a small performance drop ($97.75\\%$ to $94.90\\%$) when switching oracles. This gap arises because different oracles (ViennaRNA vs. RibonanzaNet) rely on distinct methodologies (physics-based MFE vs. deep learning) and thus model the energy landscape differently.
>
> However, CocoRNA’s generalization remains remarkably strong ($94.90\\%$). We attribute this robustness primarily to the enhanced exploration enabled by our multi-agent architecture. The cooperative, parallel exploration by multiple agents introduces significant diversity into the search process. This allows CocoRNA to avoid overfitting to the idiosyncrasies of the training oracle and identify sequences that are robustly stable (i.e., reside in deeper/broader energy minima) across different energy models.

---

### Author Response · Authors · 2025-12-02
**General Response to the Area Chair and Reviewers**

We sincerely thank the Area Chair for their time and for overseeing the review process for submission #12354.

We are encouraged by the positive assessments from the reviewers, who highlighted our work as having “**impressive empirical performance**” (`DtVm`), being “the **first multi-agent framework for RNA inverse design** with a solid theoretical foundation” (`Aczb`), and offering “a **fresh perspective** on tackling complex biological sequence design problems” (`EEwV`). We appreciate all critical feedback, which we have addressed through extensive new experiments and clarifications summarized below.

---

**Novelty and the MARL contribution (addressing `DtVm` W3, `EEwV` W3)**

A central contribution of our work is the novel formulation of RNA design as a cooperative MARL problem. This is not merely an application of existing techniques, but required specific methodological innovations to address the intractable search space and rugged energy landscape of RNA.

1. We introduced novel **decomposition schemes** (position- and structure-based) which our ablations prove are critical for performance (**App F.2**).
2. We demonstrated that the MARL architecture, specifically the CTDE framework, enables agents with local views to achieve global coordination. The centralized critic provides **globally informed gradients**, guiding the team toward the global optimum. Note that this is a capability lacking in single-agent RL approaches (`EEwV` **W3**).
3. The cooperative multi-agent approach fundamentally improves **exploration and learning stability**, addressing delayed rewards without relying on reward engineering (`PR76` **W3**).

---

**Framework flexibility and generalization**

We conducted several new experiments demonstrating that CocoRNA is flexible, oracle-agnostic, and generalizes well beyond the training distribution.

- For `PR76` **W1/Q3**, we implemented a **matrix-based variant** of CocoRNA that naturally handles **pseudoknots**. This variant not only maintained near-SOTA performance on Rfam (96.95%) but also significantly outperformed the relevant baseline (antaRNA) on pseudoknotted structures by **over 28 percentage points** (92.67% vs 64.33%).
- For `EEwV` **W1,** `PR76` **W5/W6**, we demonstrated that CocoRNA can readily incorporate **additional objectives** (GC content, MFE) via reward shaping. This allows precise control over sequence properties. This addresses the GC-bias noted by Reviewer `PR76` without compromising structural accuracy.
- To demonstrate zero-shot generalization to long sequences concerned by the `Aczb` **W3**, and `PR76` **Q4**, we tested the *original* model (**without retraining**) on sequences up to 971nt (over 2x longer than training data) using a sliding window. CocoRNA achieved a **92.4%** success rate, highlighting strong zero-shot generalization.

---

**Comprehensive benchmarking (`DtVm` W1, `PR76` W2/W4/W7, `Aczb` W2)**

We significantly expanded our evaluation to include new baselines and challenging datasets.

- We added comparisons with libLEARNA and RNAPond, and tested on the challenging Eterna100 dataset. CocoRNA maintains SOTA performance across both Rfam (97.75%) and Eterna100 (69.6%).
- We constructed **four family-specific test sets**. The analysis revealed that design difficulty is family-dependent, with Ribozymes being the most challenging due to a more rugged optimization landscape. CocoRNA still achieves 94.0% (30s) and 98.8% (600s) on this difficult family.

---

We sincerely thank the Area Chair and all reviewers for their time and efforts. We believe these clarifications and the extensive new experiments decisively address the reviewers’ concerns. We are confident the final manuscript will be a strong contribution to ICLR.

Sincerely,

The Authors of Submission #12354

---

### Meta-Review · Area_Chair_hzQs · 2026-01-06

**Summary:**

The reviewers’ reservations centered on the comprehensiveness of the benchmarking, the biological scope of the methodology, and the robustness of the reward mechanism. Specifically, Reviewers Aczb and PR76 argued that the empirical evaluation was insufficient due to the omission of critical baselines such as libLEARNA and OmniGenome, and the lack of stress-testing on challenging datasets like Eterna100 or long sequences (>500nt). A significant technical critique from Reviewer **PR76** concerned the framework's reliance on dot-bracket representation, which precludes the design of pseudoknots. Reviewer **EEwV** questioned the validity of optimizing solely for secondary structure using thermodynamic oracles (ViennaRNA) without addressing 3D conformational constraints. Additionally, concerns regarding oracle overfitting and uncontrolled biases were raised.

**Reviewer Concerns:**

The rebuttal decisively resolved the empirical and methodological deficits, particularly the concerns regarding benchmarking scope and structural limitations. The authors satisfied the rigorous evaluation demands of Reviewers Aczb, DtVm, and PR76 by integrating the Eterna100 dataset, comparing against stronger baselines like libLEARNA and OmniGenome, and demonstrating successful zero-shot generalization on sequences exceeding 500 nucleotides. They neutralized Reviewer PR76’s technical critique regarding the dot-bracket restriction by implementing a matrix-based variant that achieves SOTA on pseudoknotted structures, while simultaneously proving that reward shaping can correct GC-content biases. However, the fundamental objection from Reviewer EEwV that optimizing for secondary structure is biologically inadequate without explicit 3D modeling remains; while the authors provided a defensible justification based on the scarcity of 3D data and the standard role of 2D topology in the field, the framework remains a 2D-focused tool and does not inherently solve the reviewer's demand for 3D functional validation.

**Reviewer Scores:**

The final scoring landscape would likely solidify at 8, 4, 4, and 2 due to the distinction between empirical patchwork and fundamental satisfaction. Reviewer DtVm would maintain their strong support (8), as their request for Eterna100 benchmarks was fully met, confirming the paper's empirical robustness. Reviewer EEwV would remain anchored at 2, as the objection was philosophical—rejecting the biological validity of pure 2D folding optimization—which no amount of 2D-based reward shaping could satisfy. Reviewers Aczb and PR76 would both likely hold at 4.

---

### Decision · Program_Chairs · 2026-01-26

Reject